# COUNTERFACTUAL REASONING FOR RETRIEVAL-AUGMENTED GENERATION

**Huaiyu Qin, Chunyu Wei,**[*] **Yueguo Chen, Yunhai Wang**
Renmin University of China
Beijing, China
weichunyu@ruc.edu.cn

## ABSTRACT

While Retrieval-Augmented Generation (RAG) has advanced knowledge-intensive tasks, we identify a fundamental vulnerability: the Correlation Trap. Existing systems cannot distinguish causally decisive evidence from overwhelmingly correlated yet misleading information, leading to systematic failures. We introduce Counterfactual RAG (CF-RAG), a new framework that operationalizes causal reasoning to overcome this limitation. CF-RAG systematically generates and evaluates counterfactual queries to identify causally relevant distinctions, and employs a parallel arbitration mechanism to reconcile conflicting evidence without interference. On challenging benchmarks, CF-RAG substantially improves robustness against the Correlation Trap, achieving state-of-the-art performance while maintaining comparable efficiency to standard RAG models.[1]

## 1 INTRODUCTION

Retrieval-Augmented Generation (RAG) has revolutionized how Large Language Models (LLMs) access external knowledge, dramatically improving factual accuracy and reducing hallucinations Lewis et al. (2020); Guu et al. (2020); Borgeaud et al. (2022). By grounding responses in retrieved documents, RAG systems have become indispensable for knowledge-intensive tasks ranging from question answering to dialogue systems Izacard & Grave (2021); Karpukhin et al. (2020). Recent advances have introduced self-reflection mechanisms Asai et al. (2023), iterative refinement Yan et al. (2024), and parallel reasoning paths Wang et al. (2024c), pushing performance boundaries on standard benchmarks.

Yet these systems harbor a fundamental vulnerability: they cannot distinguish between evidence that *causally determines* an answer and distractors that are merely *correlated*. Consider the query *"Who is the lead actor in The Dark Knight?"* Conventional RAG systems retrieve documents praising Heath Ledger's Oscar-winning portrayal of the Joker, which creates overwhelming correlational signals that mislead models into identifying him as the lead, rather than Christian Bale (Figure 1). We term this systematic failure the **Correlation Trap**.

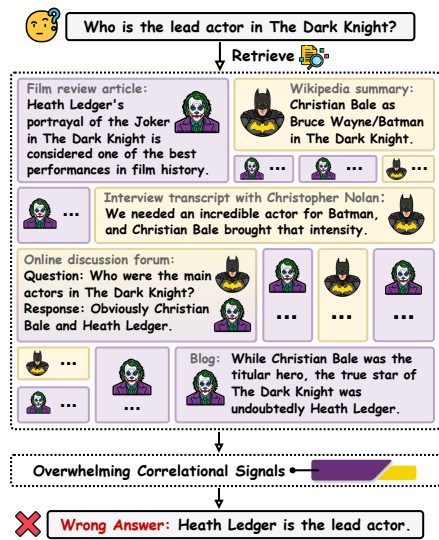

Figure 1: **The Correlation Trap.** strong correlational signals that overshadow the causal evidence identifying Christian Bale as the protagonist.

This vulnerability reveals a deeper limitation: existing RAG systems lack causal reasoning about *why* evidence supports specific answers. While correlation-based retrieval excels at finding topically relevant documents, it fails when correctness depends on understanding causal relationships, such

---

[*]Corresponding author: weichunyu@ruc.edu.cn

[1]The code is available on https://github.com/CF-RAG/CF-RAG

as distinguishing protagonists from antagonists, primary authors from contributors, or lead actors from supporting cast. The challenge is not merely retrieving more documents but fundamentally rethinking how systems reason about evidence.

Bridging the gap between correlation and causation in RAG faces two critical obstacles:

- **Causal Distinction:** Which conceptual boundaries matter for a given query? For our example, the protagonist-antagonist distinction is causally decisive, while the British-American actor distinction is incidental. Without supervision, systems must autonomously discover these critical boundaries from an exponentially large space of possible distinctions, which is a challenge current methods cannot address.
- **Evidence Reconciliation:** How should systems handle conflicting signals in retrieved evidence? When documents simultaneously praise Bale's lead performance and Ledger's scene-stealing portrayal, joint processing causes interference (stronger correlational signals overshadow weaker causal ones), while separate processing loses comparative context. The system must maintain distinct reasoning paths while enabling meaningful cross-validation.

We propose **CF-RAG (Counterfactual RAG)**, a framework that transforms correlation-based retrieval into causation-driven reasoning through systematic counterfactual testing. Our key insight: causal evidence uniquely supports the original query over plausible alternatives, while correlational evidence fails this discrimination test. CF-RAG introduces two synergistic mechanisms:

**Counterfactual Exploration** systematically generates alternative hypotheses that probe conceptual boundaries. For our example query, CF-RAG explores role variations ("Who played the villain?"), categorical shifts ("Who directed the film?"), and temporal boundaries ("Who starred in Batman Begins?"). By retrieving evidence for both original and counterfactual queries, we create a dialectical evidence space revealing which distinctions are causally decisive.

**Parallel Arbitration** maintains multiple reasoning paths while enabling rigorous comparison. CF-RAG constructs diverse evidence subsets through stratified sampling, generates parallel answer drafts, and evaluates them using dual scoring: *internal coherence* (alignment with supporting evidence) and *causal discrimination* (whether evidence uniquely supports the original over counterfactuals). This architecture prevents interference from conflicting signals while identifying evidence that makes answers necessary rather than merely plausible.

Our contributions are threefold:

1. We formalize causal reasoning in RAG through counterfactual testing, providing principled methods for distinguishing causation from correlation in evidence evaluation.
2. CF-RAG operationalizes causal reasoning through counterfactual exploration and parallel arbitration, maintaining computational tractability while achieving robust performance.
3. We achieve state-of-the-art results on challenging benchmarks, with substantial improvements in robustness against misleading correlations while maintaining efficiency comparable to baselines.

## 2 RELATED WORK

**Retrieval-Augmented Generation.** RAG has become the dominant paradigm for grounding LLM responses in external knowledge, substantially reducing hallucinations (Lewis et al., 2020; Gao et al., 2023; Sharma, 2025; Khandelwal et al., 2020). Recent advances target different aspects of the retrieval-generation pipeline: adaptive retrieval methods learn when and what to retrieve (Jiang et al., 2023b; Ma et al., 2023; Chen et al., 2023; Schick et al., 2024), context utilization improvements include chain-of-thought retrieval (Yu et al., 2023), hierarchical organization (Sarthi et al., 2024), and compression techniques (Xu et al., 2023; Kim et al., 2024; Yoran et al., 2023; Wang et al., 2023; Baek et al., 2023). Robustness mechanisms address noisy retrieved content through various strategies. Self-RAG introduces reflection tokens to critique passages (Asai et al., 2024). CRAG employs lightweight evaluators for filtering (Yan et al., 2024), RAFT trains models to ignore distractors (Zhang et al., 2024), and SAIL (Luo et al., 2023) filters irrelevant content using web-search-tuned models. While these methods improve resilience to noise (Shi et al., 2023; Liu et al., 2024) and explore architectural innovations (Xia et al., 2024; Wang et al., 2024b), they fundamentally rely on correlational signals, optimizing semantic relevance without modeling causal

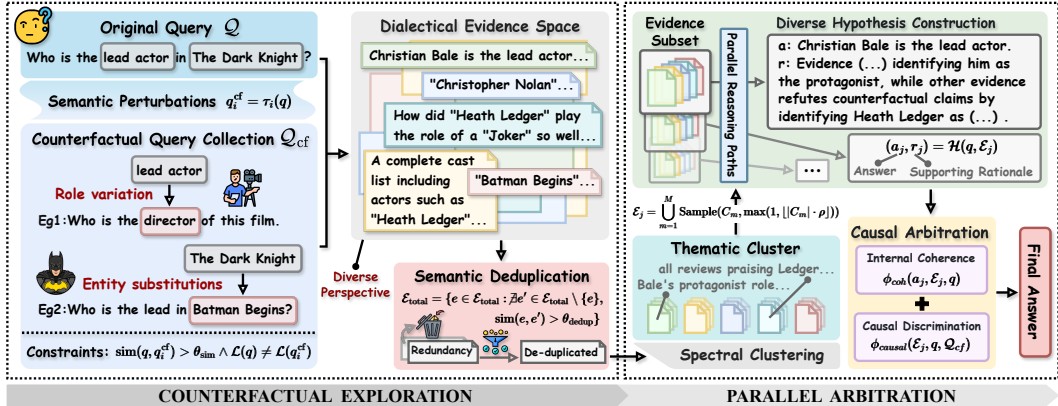

Figure 2: **CF-RAG Framework Overview.** (1) **Counterfactual Exploration**, which generates counterfactual queries to build a dialectical evidence space, and (2) **Parallel Arbitration**, which constructs and evaluates multiple hypotheses in parallel to identify the causally supported answer.

relationships between evidence and answers. CF-RAG addresses this gap by introducing causal reasoning directly into evidence selection through counterfactual testing.

**Counterfactual Reasoning in NLP.** Counterfactual reasoning has gained traction for improving model robustness and interpretability (Wang et al., 2024a; Singh et al., 2024). Counterfactual explanation methods generate minimally-edited inputs that flip model predictions, revealing decision boundaries. Polyjuice enables controlled perturbations (Wu et al., 2021) while FIZLE achieves zero-shot generation (Bhattacharjee et al., 2024; McAleese & Keane, 2024). Counterfactual data augmentation enhances training robustness: Kaushik et al. (2020) demonstrated that human-generated counterfactuals improve out-of-distribution generalization, with subsequent work automating generation at scale (Balashankar et al., 2023). However, these approaches treat counterfactuals as either post-hoc analysis tools or offline training resources. CF-RAG pioneers integrating generation and evaluation directly into the RAG inference pipeline for real-time causal verification.

## 3 PRELIMINARIES

**Definition 1 (Retrieval-Augmented Generation)** *Given a query space $\mathcal{Q}$, answer space $\mathcal{A}$, document corpus $\mathcal{D} = \{d_1, \ldots, d_N\}$, and retrieval function $\mathcal{R} : \mathcal{Q} \times \mathcal{D} \to 2^{\mathcal{D}}$, standard RAG seeks: $a^* = \arg\max_{a \in \mathcal{A}} P(a \mid q, \mathcal{R}(q, \mathcal{D}))$, where $P(a \mid q, \mathcal{R}(q, \mathcal{D}))$ represents the posterior probability of answer $a$ conditioned on query $q$ and retrieved evidence.*

This formulation suffers from the *correlation trap*: when spurious correlations dominate retrieved evidence, systems conflate co-occurrence with causal support, motivating our causal reformulation:

**Definition 2 (Causal RAG)** *A causal RAG system enforces counterfactual robustness by requiring that generated answers satisfy discriminative causality. For query $q$ and counterfactual set $\mathcal{Q}_{cf} \subset \mathcal{Q}$:*

$$a^* = \arg\max_{a \in \mathcal{A}} P(a \mid q, \mathcal{D}) \cdot \mathbb{I}\left[\forall q' \in \mathcal{Q}_{cf} : \Phi(a, q, \mathcal{D}) > \Phi(a, q', \mathcal{D})\right] \tag{1}$$

*where $\Phi : \mathcal{A} \times \mathcal{Q} \times \mathcal{D} \to \mathbb{R}$ quantifies causal support, which is the evidence that makes answer $a$ necessary for query $q$ rather than merely consistent with it.*

## 4 METHODOLOGY

CF-RAG transforms correlation-based retrieval into causation-driven reasoning through two synergistic mechanisms that directly address the challenges identified in our introduction: **Counterfactual Exploration** for discovering causal boundaries, and **Parallel Arbitration** for reconciling conflicting evidence. The architecture of CF-RAG is illustrated in Figure 2.

### 4.1 Counterfactual Exploration

Counterfactual Exploration addresses the challenge of causal distinction: how a system can identify which conceptual boundaries causally matter for a query. Our key insight is that causal evidence exhibits *discriminative selectivity*. It will strongly support the original query while showing measurably weaker support for semantically similar alternatives. Conversely, spurious correlations exhibit *indiscriminate activation*, supporting multiple related queries similarly.

#### 4.1.1 Structured Counterfactual Query Generation

Given an original query $q$, we systematically generate counterfactual queries that probe different semantic dimensions through controlled perturbations. This creates a comprehensive stress test for evidence. If a document equally supports both the original query and semantically related alternatives, it likely contains correlational rather than causal information about the queried relationship.

**Semantic Transformation Framework.** We formalize counterfactual generation as a structured transformation process. Let $\mathcal{Q}$ be the query space and $\mathcal{T} = \{\tau_1, \tau_2, \ldots, \tau_n\}$ denote a set of transformation functions where each $\tau_i : \mathcal{Q} \to \mathcal{Q}$ modifies a specific semantic dimension. For a given query $q$, we generate counterfactual queries as: $\mathcal{Q}_{\text{cf}} = \{q_i^{\text{cf}} = \tau_i(q) : i \in [1, n], \mathcal{V}(q, q_i^{\text{cf}}) = \texttt{True}\}$, where $\mathcal{V}(q, q')$ is a validation function ensuring counterfactual quality:

$$\mathcal{V}(q, q') = \begin{cases} \texttt{True} & \text{if } \text{sim}_{\text{sem}}(q, q') > \theta_{\text{sim}} \wedge \mathcal{L}(q) \neq \mathcal{L}(q') \\ \texttt{False} & \text{otherwise} \end{cases} \tag{2}$$

The constraint $\text{sim}_{\text{sem}}(q, q') > \theta_{\text{sim}}$ ensures topical coherence, while $\mathcal{L}(q) \neq \mathcal{L}(q')$ guarantees different ground-truth answers, creating the necessary tension for causal discrimination.

**Transformation Taxonomy.** We employ five categories of semantic transformations, each targeting different aspects of causal relationships: (1) `Role-based` ($\tau_{\text{role}}$) alter functional roles while maintaining domain relevance; (2) `Temporal-shift` ($\tau_{\text{time}}$) probe chronological causality through temporal modifications; (3) `Entity-substitution` ($\tau_{\text{entity}}$) test entity-specific versus general patterns; (4) `Categorical-inversion` ($\tau_{\text{cat}}$) explore opposite categorical boundaries; (5) `Scope-modification` ($\tau_{\text{scope}}$) adjust query breadth and specificity. Detailed descriptions and examples along with the prompt template for transformation generation are provided in Appendix F.

**Adaptive Counterfactual Selection.** Not all transformations are equally informative for every query. We introduce an adaptive selection mechanism that prioritizes counterfactuals based on their potential to reveal causal distinctions. Given a query $q$ and candidate counterfactual set $\mathcal{Q}_{\text{cand}}$, we compute an informativeness score for each candidate:

$$\text{Info}(q, q') = \alpha \cdot \text{Div}_{\text{sem}}(q, q') + \beta \cdot \text{Div}_{\text{ans}}(q, q') + \gamma \cdot \text{Rel}_{\text{dom}}(q, q') \tag{3}$$

where $\text{Div}_{\text{sem}}(q, q') = 1 - \text{sim}_{\text{sem}}(q, q')$ measures semantic divergence; $\text{Div}_{\text{ans}}(q, q') = \text{dist}(\mathcal{A}_q, \mathcal{A}_{q'})$ measures expected answer space distance; and $\text{Rel}_{\text{dom}}(q, q') = \text{sim}_{\text{domain}}(q, q')$ measures domain relevance. We select the top-$K$ counterfactuals: $\mathcal{Q}_{\text{cf}} = \text{TopK}(\mathcal{Q}_{\text{cand}}, \text{Info}(\cdot, \cdot), K)$.

#### 4.1.2 Dialectical Evidence Retrieval

Traditional RAG systems retrieve evidence exclusively for the original query, creating an echo chamber where correlational patterns reinforce themselves without challenge. CF-RAG instead constructs a *dialectical evidence space* that deliberately incorporates potentially contradictory perspectives, forcing the system to confront and resolve conflicting signals.

**Multi-Query Evidence Aggregation.** We perform retrieval across all queries in the expanded query set $\{q\} \cup \mathcal{Q}_{\text{cf}}$: $\mathcal{E}_{\text{total}} = \mathcal{R}(q, \mathcal{D}) \cup \bigcup_{i=1}^{K} \mathcal{R}(q_i^{\text{cf}}, \mathcal{D})$, where $\mathcal{R} : \mathcal{Q} \times \mathcal{D} \to 2^{\mathcal{D}}$ is our retrieval function. This dialectical retrieval simultaneously exposes spurious correlations by revealing documents that support multiple related queries with similar strength, generates contrastive signals that provide negative evidence against incorrect interpretations, and ensures comprehensive coverage by mitigating query formulation biases that might otherwise overlook relevant perspectives.

**Semantic Deduplication and Quality Filtering.** To address redundancy and low-quality content in $\mathcal{E}_{\text{total}}$, we apply sequential filtering: first removing near-duplicate documents where $\text{sim}_{\text{emb}}(e, e') > \theta_{\text{dedup}}$ (Equation 4), then retaining only documents that satisfy both quality thresholds $\mathcal{Q}(e) > \theta_{\text{quality}}$

and relevance requirements $\max_{q' \in \{q\} \cup \mathcal{Q}_{\text{cf}}} s(q', e) > \theta_{\text{rel}}$ (Equation 5), where $\mathcal{Q}(e)$ measures coherence, informativeness, and factual reliability.

$$\mathcal{E}_{\text{dedup}} = \{e \in \mathcal{E}_{\text{total}} : \nexists e' \in \mathcal{E}_{\text{total}} \setminus \{e\}, \text{sim}_{\text{emb}}(e, e') > \theta_{\text{dedup}}\} \tag{4}$$

$$\mathcal{E}_{\text{filtered}} = \{e \in \mathcal{E}_{\text{dedup}} : \mathcal{Q}(e) > \theta_{\text{quality}} \wedge \max_{q' \in \{q\} \cup \mathcal{Q}_{\text{cf}}} s(q', e) > \theta_{\text{rel}}\} \tag{5}$$

## 4.2 PARALLEL ARBITRATION

Parallel Arbitration addresses the challenge of evidence reconciliation: how can a system validate competing interpretations when evidence contains conflicting signals? Traditional approaches either process all evidence jointly (leading to interference between conflicting signals) or separately (losing comparative context). Our solution maintains multiple reasoning paths in parallel while enabling rigorous cross-validation through causal arbitration.

### 4.2.1 EVIDENCE STRATIFICATION AND HYPOTHESIS CONSTRUCTION

When heterogeneous evidence is processed jointly, strong correlational signals can overshadow weaker but more causally relevant evidence. To prevent this interference while maintaining diversity, we partition the evidence space and construct multiple hypotheses from different perspectives.

**Thematic Clustering.** We partition $\mathcal{E}_{\text{filtered}}$ into $M$ thematic clusters using spectral clustering on document embeddings. Let $\mathbf{E} \in \mathbb{R}^{|\mathcal{E}_{\text{filtered}}| \times d}$ be the matrix of document embeddings. We construct an affinity matrix: $W_{ij} = \exp\left(-\frac{\|\mathbf{e}_i - \mathbf{e}_j\|^2}{2\sigma^2}\right)$. The normalized Laplacian is computed as $\mathcal{L} = I - D^{-1/2} W D^{-1/2}$, where $D$ is the degree matrix. We perform eigendecomposition and use the first $M$ eigenvectors for $K$-means clustering: $\mathcal{C} = \{C_1, C_2, \ldots, C_M\} = \text{KMeans}(\mathbf{V}_M, M)$, where $\mathbf{V}_M$ contains the first $M$ eigenvectors of $\mathcal{L}$.

**Stratified Evidence Sampling.** For each of $P$ parallel reasoning paths, we construct a diverse evidence subset through stratified sampling that ensures representation from all thematic perspectives: $\mathcal{E}_j = \bigcup_{m=1}^{M} \text{WeightedSample}(C_m, n_{jm})$, where the sample size from each cluster is determined by:

$$n_{jm} = \max\left(1, \lfloor |C_m| \cdot \rho \cdot w_{jm} \rfloor\right) \tag{6}$$

The weight $w_{jm}$ introduces controlled randomness to ensure diversity across reasoning paths:

$$w_{jm} = \text{Softmax}\left(\frac{\log(\text{Uniform}(0, 1)) + \mu_m}{\tau}\right) \tag{7}$$

where $\mu_m$ is a cluster-specific bias term and $\tau$ controls the randomness temperature.

**Multi-Perspective Hypothesis Generation.** Each evidence subset generates a hypothesis consisting of both an answer and its supporting rationale:

$$(a_j, r_j) = \mathcal{H}(q, \mathcal{E}_j) = \text{LLM}_{\text{generator}}(\text{Prompt}_{\text{hypothesis}}(q, \mathcal{E}_j)) \tag{8}$$

Please refer to Appendix G for the carefully designed prompt structure. This parallel architecture enables the system to explore multiple interpretations simultaneously, allowing different hypotheses to emerge from different evidence perspectives while preventing mutual interference.

### 4.2.2 CAUSAL ARBITRATION MECHANISM

The core innovation of CF-RAG lies in its arbitration mechanism, which transcends traditional confidence-based selection by incorporating causal reasoning principles. Rather than relying solely on internal coherence metrics, we evaluate each hypothesis through a multi-criterion system that captures both evidence faithfulness and causal discriminative power.

**Internal Coherence Scoring.** The coherence score $\phi_{\text{coh}}$ measures answer-evidence alignment to ensure grounded responses by averaging coherence functions across supporting documents (Equation 9), where each function combines semantic similarity between answer and document embeddings with query-relevance weighted by explicit answer mentions (Equation 10).

$$\phi_{\text{coh}}(a_j, \mathcal{E}_j, q) = \frac{1}{|\mathcal{E}_j|} \sum_{e \in \mathcal{E}_j} \text{CoherenceFunction}(a_j, e, q) \tag{9}$$

$$\text{CoherenceFunction}(a, e, q) = \lambda_1 \cdot \text{sim}_{\text{sem}}(\text{Enc}(a), \text{Enc}(e)) + \lambda_2 \cdot s(q, e) \cdot \text{Mention}(a, e) \tag{10}$$

where $\text{Enc} : \mathcal{A} \cup \mathcal{D} \rightarrow \mathbb{R}^d$ maps answers and documents to embeddings, $s(q, e)$ measures query-document relevance, and $\text{Mention}(a, e) \in \{0, 1\}$ indicates explicit answer mention.

**Causal Discrimination Scoring.** Causal score $\phi_{\text{causal}}$ quantifies whether evidence supports the original query over counterfactual alternatives—measuring necessity rather than mere consistency:

$$\phi_{\text{causal}}(\mathcal{E}_j, q, \mathcal{Q}_{\text{cf}}) = \frac{1}{|\mathcal{E}_j|} \sum_{e \in \mathcal{E}_j} \left[ s(q, e) - \max_{q' \in \mathcal{Q}_{\text{cf}}} s(q', e) \right] \tag{11}$$

This formulation captures the key insight of counterfactual reasoning: causal evidence should exhibit differential support across related queries. A positive causal score indicates that evidence provides stronger support for the original query than for any counterfactual, suggesting genuine causal relevance rather than spurious correlation.

**Multi-Criteria Scoring Integration.** We extend the basic dual-criterion approach to incorporate additional robustness measures. The comprehensive scoring function becomes: $\Psi_j = \sum_{k=1}^{4} w_k \cdot \phi_k^{(j)}$, where the four criteria are:

$\phi_1^{(j)} = \phi_{\text{coh}}(a_j, \mathcal{E}_j, q)$    (Internal Coherence)

$\phi_2^{(j)} = \phi_{\text{causal}}(\mathcal{E}_j, q, \mathcal{Q}_{\text{cf}})$    (Causal Discrimination)

$\phi_3^{(j)} = \text{Confidence}(a_j, r_j)$    (Generation Confidence)

$\phi_4^{(j)} = \text{Specificity}(a_j, q)$    (Answer Specificity)

The confidence score measures the model's certainty in its generated response:

$$\text{Confidence}(a, r) = \frac{1}{|r|} \sum_{t=1}^{|r|} \log P(t_r | t_{r-1}, \ldots, t_1, a, q)$$

The specificity score rewards answers that directly address the query rather than providing overly general responses:

$$\text{Specificity}(a, q) = 1 - \frac{\text{Entropy}(\text{KeywordDist}(a))}{\log |\text{Vocabulary}(a)|}$$

**Algorithm 1: Pseudo code of CF-RAG**

---

**Require:** Query $q$, Corpus $\mathcal{D}$, Parameters $K$, $M$, $P$, $\lambda$
**Ensure:** Final answer $a_{\text{final}}$ with rationale $r_{\text{final}}$
1:   **# PHASE 1: COUNTERFACTUAL EXPLORATION**
2:   $\mathcal{Q}_{\text{cf}} \leftarrow$ **GenerateCounterfactuals**$(q, K)$
3:   $\mathcal{E}_{\text{dialectical}} \leftarrow \mathcal{R}(q, \mathcal{D}) \cup \bigcup_{q' \in \mathcal{Q}_{\text{cf}}} \mathcal{R}(q', \mathcal{D})$
4:   $\mathcal{E}_{\text{filtered}} \leftarrow$ FilterAndDeduplicate$(\mathcal{E}_{\text{dialectical}})$
5:   **# PHASE 2: PARALLEL ARBITRATION**
6:   $\mathcal{C} \leftarrow$ ClusterEvidence$(\mathcal{E}_{\text{filtered}}, M)$
7:   **for** $j = 1$ to $P$ **do**
8:      $\mathcal{E}_j \leftarrow$ **StratifiedSample**$(\mathcal{C})$
9:      $(a_j, r_j) \leftarrow$ GenerateHypothesis$(q, \mathcal{E}_j)$
10:     $\phi_{\text{coh}}^j \leftarrow$ InternalCoherence$(a_j, \mathcal{E}_j, q)$
11:     $\phi_{\text{causal}}^j \leftarrow$ **CausalDiscrimination**$(\mathcal{E}_j, q, \mathcal{Q}_{\text{cf}})$
12:     $\Psi_j \leftarrow (1 - \lambda) \cdot \phi_{\text{coh}}^j + \lambda \cdot \phi_{\text{causal}}^j$
13:   **end for**
14:   $j^* \leftarrow \arg\max_j \Psi_j$
15:   **if** HighConsensus$(\{a_j\})$ **then**
16:     $a_{\text{final}} \leftarrow a_{j^*}$
17:   **else**
18:     $a_{\text{final}} \leftarrow$ Synthesize(TopK$(\{a_j\}, 3)$)
19:   **end if**
20:   **return** $(a_{\text{final}}, r_{\text{final}})$

---

### 4.2.3 Answer Synthesis and Refinement

Beyond selecting the highest-scoring hypothesis $j^* = \arg\max_j \Psi_j$, we perform cross-hypothesis validation by measuring agreement patterns $\text{Agreement}(a) = \frac{|\{j : \text{sim}(a_j, a) > \theta_{\text{agree}}\}|}{P}$ to boost confidence for consensus answers or trigger verification for divergent ones. When multiple high-scoring hypotheses exist, we synthesize a combined response from the top-3 candidates using specialized prompts that incorporate multi-perspective evidence while maintaining coherence.

$$a_{\text{final}} = \text{Synthesize}\left(\{(a_j, \Psi_j)\}_{j \in \text{TopK}(\Psi, 3)}, q, \mathcal{E}_{\text{filtered}}\right) \tag{12}$$

Algorithm 1 presents the complete CF-RAG framework.

## 5 Theoretical Analysis

CF-RAG escapes the correlation trap through causal discrimination that remains robust even when spurious evidence vastly outnumbers causal evidence.

**Lemma 1 (Counterfactual Discriminability)** *For CF-RAG to distinguish causal from spurious evidence, $\mathcal{Q}_{cf}$ must contain at least one query $q'$ such that spurious evidence $\mathcal{E}_{spurious}$ exhibits comparable support for both queries:*    $\exists q' \in \mathcal{Q}_{cf} : \forall e \in \mathcal{E}_{spurious}, \quad |s(q, e) - s(q', e)| < \epsilon.$

**Lemma 2 (Volume Invariance)** *The causal discrimination score $\phi_{causal}(\mathcal{E}, q, \mathcal{Q}_{cf})$ is invariant under spurious evidence scaling:*    $\phi_{causal}(\alpha \cdot \mathcal{E}_{spurious} \cup \mathcal{E}_{causal}, q, \mathcal{Q}_{cf}) = \phi_{causal}(\mathcal{E}_{spurious} \cup \mathcal{E}_{causal}, q, \mathcal{Q}_{cf})$ *for any scaling factor $\alpha > 0$.*

Together, these lemmas ensure spurious evidence detection (Lemma 1) remains effective regardless of volume (Lemma 2), enabling our main result:

**Theorem 1 (Escape from the Correlation Trap)** *Let $q$ be a query with ground truth answer $a^*$ supported by causal evidence $\mathcal{E}_{causal}$ and spurious answer $a'$ supported by correlated evidence $\mathcal{E}_{spurious}$. If: **(C1)** $\forall e \in \mathcal{E}_{causal}, \forall q' \in \mathcal{Q}_{cf} : s(q,e) - s(q',e) \geq \delta$; **(C2)** By Lemma 1: $\exists q' \in \mathcal{Q}_{cf}$ such that $\forall e \in \mathcal{E}_{spurious} : |s(q,e) - s(q',e)| < \epsilon$; **(C3)** $\delta > \epsilon$.*

*Then CF-RAG with appropriate $\lambda$ selects $a^*$ over $a'$ regardless of volume ratio $|\mathcal{E}_{spurious}|/|\mathcal{E}_{causal}|$, while standard RAG fails when this ratio exceeds $\bar{s}_{causal}/\bar{s}_{spurious}$.*

The causal discrimination condition (C1) ensures that genuine causal evidence maintains its discriminative power under counterfactual testing. The spurious non-discrimination condition (C2) directly invokes Lemma 1 to guarantee that spurious correlations can be detected. The separation condition (C3) ensures reliable discrimination in practice. Proofs are provided in Appendix A.

# 6 EXPERIMENTS

## 6.1 EXPERIMENTAL SETUP

**Datasets.** We evaluate on five challenging benchmarks that test different aspects of causal reasoning under retrieval-augmented generation. These include multi-hop QA datasets **HotpotQA** Yang et al. (2018) and **MusiQue** Trivedi et al. (2022), which require synthesizing evidence across multiple documents while avoiding distractors; **TriviaQA** Joshi et al. (2017), featuring compositionally complex questions with subtle spurious correlations; **PopQA** Mallen et al. (2023), focusing on long-tail entities where models must rely on retrieved rather than parametric knowledge; and **PubHealth** Zhang et al. (2023), a medical claim verification task demanding high factual precision. Each dataset presents unique challenges for disambiguating causally relevant evidence from misleading correlations. See Appendix C.1 for detailed dataset characteristics and preprocessing pipeline.

**Baselines.** We compare against three categories of systems: (i) *Zero-shot LLMs* without retrieval (Llama-3-8B, Llama-2-7B, Alpaca-7B); (ii) *Standard RAG* augmenting these models with conventional retrieval; and (iii) *Advanced RAG frameworks* including CRAG (Yan et al., 2024), Self-RAG (Asai et al., 2024), Self-CRAG (Yan et al., 2024), and Speculative-RAG (Wang et al., 2024c). Additionally, we evaluate tool-augmented approaches: Toolformer (Schick et al., 2024) and SAIL (Luo et al., 2023). See Appendix B.4 for implementation details.

**Implementation.** CF-RAG uses Llama-3-8B-Instruct and Llama-2-7B as base models for fair comparison. Default hyperparameters: $N=3$ counterfactual queries, $K=4$ evidence clusters, $M=3$ parallel drafts, and causal weight $\lambda=0.4$. All models use identical retrieval corpora and are evaluated using Exact Match (EM) scores. See Appendix B for complete details.

## 6.2 MAIN RESULTS

Table 1 presents our main experimental findings. CF-RAG demonstrates substantial improvements across all benchmarks, with the most dramatic gains on complex reasoning tasks:

**Multi-hop Reasoning Excellence.** On HotpotQA, CF-RAG (Llama-3-8B) achieves 88.58% EM, an absolute improvement of **+80.8%** over the strongest baseline. This remarkable gain—nearly doubling performance—validates our hypothesis that causal reasoning is essential for navigating complex evidence chains. The improvement on MusiQue (+63.2%) further confirms this pattern.

**Robust Performance Across Domains.** Beyond multi-hop tasks, CF-RAG maintains strong performance on single-hop retrieval (TriviaQA: 81.02%), long-tail entities (PopQA: 73.57%), and medical claims (PubHealth: 83.36%). This consistency suggests that causal reasoning benefits extend beyond specific task types to fundamentally improve evidence evaluation.

**Model-Agnostic Improvements.** Both Llama-2-7b and Llama-3-8B backbones show substantial gains (+21.0% and +31.9% average improvement respectively), demonstrating that CF-RAG's benefits are not tied to specific model architectures.

Table 1: **Performance comparison across five QA benchmarks.** All scores are Exact Match (%). The best results from baseline are underlined.

| Method | HotpotQA | TriviaQA | PopQA | MusiQue | PubHealth | Avg. |
|---|---|---|---|---|---|---|
| *Zero-Shot LLM (No Retrieval)* | | | | | | |
| Llama-2-7b | 18.21 | 18.53 | 13.82 | 8.12 | 32.09 | 18.15 |
| Llama-3-8B-Instruct | 23.45 | 28.17 | 20.31 | 14.30 | 44.27 | 26.10 |
| Alpaca-7B | 23.83 | 26.20 | 23.60 | 12.90 | 49.80 | 27.27 |
| *Standard RAG* | | | | | | |
| Llama-2-7b | 27.72 | 32.52 | 38.20 | 15.84 | 38.10 | 30.48 |
| Llama-3-8B-Instruct | 36.04 | 31.12 | 42.46 | 21.41 | 48.58 | 35.92 |
| Alpaca-7B | 31.29 | 34.20 | 46.70 | 19.90 | 40.20 | 34.46 |
| *Instruction-Tuned RAG* | | | | | | |
| SAIL | – | – | – | – | 69.20 | – |
| Toolformer | – | 48.80 | – | – | – | – |
| *Advanced RAG Frameworks* | | | | | | |
| CRAG (Llama-2-7b) | 25.24 | 60.77 | 55.02 | – | 60.18 | – |
| Self-RAG (Llama-2-7b) | 28.49 | 64.39 | 53.97 | 20.58 | 73.19 | 48.12 |
| Self-CRAG (Llama-2-7b) | 34.44 | 63.72 | 60.98 | – | 74.80 | – |
| Speculative-RAG (Mistral-7b) | 49.00 | 74.24 | 57.54 | 31.57 | 76.60 | 57.79 |
| Speculative-RAG (Llama-2-7b) | 47.90 | 75.37 | 56.21 | 33.45 | 75.29 | 57.64 |
| **CF-RAG (Llama-2-7b)** | **79.29** (↑61.8%) | **76.15** (↑1.0%) | **67.22** (↑10.2%) | **48.78** (↑45.8%) | **78.24** (↑2.1%) | **69.94** (↑21.0%) |
| **CF-RAG (Llama-3-8B)** | **88.58** (↑80.8%) | **81.02** (↑7.5%) | **73.57** (↑20.6%) | **54.59** (↑63.2%) | **83.36** (↑8.8%) | **76.22** (↑31.9%) |

## 6.3 COMPONENT ANALYSIS

To understand each component's contribution, we conduct systematic ablations on HotpotQA and PopQA. Table 2 presents the results, revealing a clear component hierarchy. Removing the causal verification mechanism ($\phi_{causal}$) causes the most severe degradation—a 17.37% drop on HotpotQA and 13.73% on PopQA—confirming that distinguishing causal from correlational evidence is CF-RAG's core innovation. The counterfactual exploration component is also critical, its removal causes 11.36% and 6.17% performance drops respectively, demonstrating dialectical evidence construction is essential for enabling causal discrimination. Even the evidence division mechanism, while contributing the smallest individual effect, yields meaningful improvements (4.84% on HotpotQA, 8.54% on PopQA) by preventing interference between conflicting signals during parallel hypothesis generation. These results validate our design: each component addresses a specific failure mode in correlation-based retrieval, and their synergistic combination enables the robust causal reasoning distinguishing CF-RAG from existing approaches.

Table 2: **Ablation study revealing component contributions.** Each column shows performance with a specific component removed, with degradation percentages in parentheses.

| Dataset | Full CF-RAG | w/o Counterfactual | w/o Evidence Division | w/o Causal Verification |
|---|---|---|---|---|
| HotpotQA | **88.58** | 78.52 (↓11.36%) | 84.29 (↓4.84%) | 73.19 (↓17.37%) |
| PopQA | **73.57** | 69.03 (↓6.17%) | 67.29 (↓8.54%) | 63.47 (↓13.73%) |

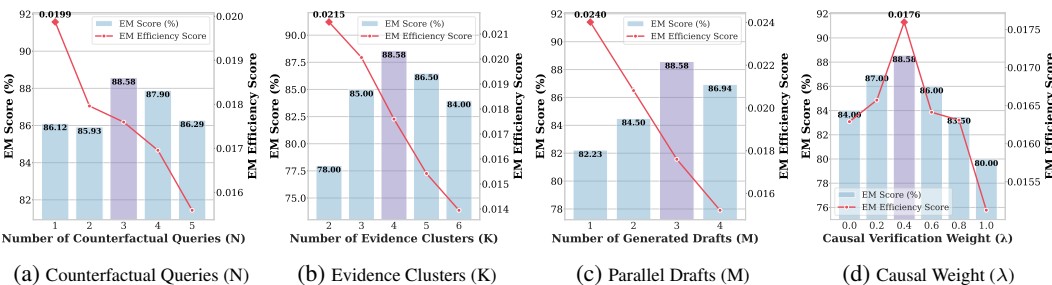

(a) Counterfactual Queries (N)  (b) Evidence Clusters (K)  (c) Parallel Drafts (M)  (d) Causal Weight (λ)

Figure 3: **Hyperparameter sensitivity analysis on HotpotQA.** The bar chart shows the EM score (accuracy) and the line chart shows the EM efficiency score (accuracy/latency).

## 6.4 PARAMETER SENSITIVITY AND EFFICIENCY ANALYSIS

We analyze CF-RAG's hyperparameter trade-offs between accuracy and computational efficiency on HotpotQA, introducing an **EM Efficiency Score** that measures accuracy per second: EM Efficiency $= \frac{\text{EM Score}/100}{\text{Latency (s)}}$. This metric reveals how CF-RAG adapts to different operational requirements (Figure 3).

**Accuracy-Efficiency Trade-offs.** The number of counterfactual queries ($N$), evidence clusters ($K$), and parallel drafts ($M$) exhibit consistent patterns: peak accuracy occurs at moderate values ($N$=3, $K$=4, $M$=3) achieving 88.58% EM, while peak efficiency emerges at minimal settings ($N$=1, $K$=2, $M$=1) for maximum throughput. These findings enable flexible deployment—quality-critical applications can maximize accuracy with higher parameters, while latency-sensitive systems can achieve faster responses with minimal settings while maintaining 80%+ accuracy.

**Causal Weight Optimization.** The causal weight $\lambda$ uniquely shows aligned optima: both accuracy and efficiency peak at $\lambda$=0.4, validating our core hypothesis that balanced synthesis of coherence and causal discrimination is essential. Pure coherence-based ($\lambda$=0) or purely causal ($\lambda$=1.0) selection underperform by 5% and 9.7% respectively, confirming that neither signal alone suffices for robust performance. This parameter requires no trade-off—the theoretically motivated balance point also maximizes practical efficiency.

## 6.5 MECHANISTIC ANALYSIS: ADDRESSING RAG FAILURE MODES

Table 3: **Performance on specific RAG failure modes.** CF-RAG's targeted mechanisms dramatically reduce error rates across all failure categories.

| Failure Mode | Description | Baseline Error | CF-RAG Error | Reduction |
|---|---|---|---|---|
| Spurious Correlation | Relies on correlated but non-causal evidence | 56.7% | 13.3% | **76.5%** |
| Evidence Extraction | Correct answer obscured by noise | 43.3% | 23.3% | **46.1%** |
| Scattered Synthesis | Fails to connect multi-hop evidence | 63.3% | 16.7% | **73.6%** |

To understand *why* CF-RAG succeeds, we analyze its performance on three documented RAG failure modes using curated challenge sets (30 queries per mode) as illustrated in Table 3. The dramatic error reductions demonstrate that CF-RAG's components directly address root causes: (1) Causal verification eliminates spurious correlations; (2) Parallel processing isolates signal from noise; (3) Counterfactual exploration bridges evidence gaps.

## 6.6 ROBUSTNESS TO ADVERSARIAL DISTRACTION

We stress-test robustness by injecting $N_{\text{inject}} \in \{2, 4, 8, 16\}$ adversarially-selected distractors into HotpotQA's retrieval corpus. These documents maximize spurious correlations: high semantic similarity but zero factual relevance.

As shown in Figure 4, baseline models degrade significantly under adversarial pressure, with accuracy dropping by 66.4% for Standard RAG and 56% for Speculative-RAG. In stark contrast, **CF-RAG** demonstrates remarkable stability, maintaining a high accuracy of **60.57%** in the most challenging setting. This resilience validates that the framework's decisions are driven by causal discrimination rather than correlation strength, ensuring reliable performance in noisy environments where other systems falter.

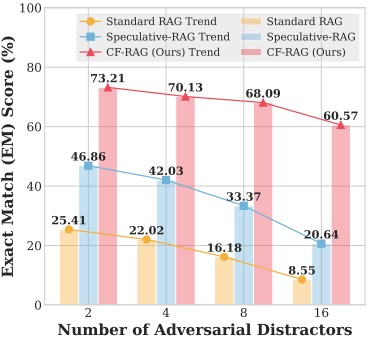

Figure 4: **Adversarial robustness.**

## 7 CONCLUSIONS AND LIMITATION

In this work, we identified and addressed a fundamental vulnerability in retrieval-augmented generation systems: the Correlation Trap, where models conflate spurious correlations with causal evidence. We introduced CF-RAG, a novel framework that transforms correlation-based retrieval into causation-driven reasoning through systematic counterfactual testing. By integrating Counterfactual

Exploration to discover causal boundaries and Parallel Arbitration to reconcile conflicting evidence, CF-RAG achieves robust discrimination between evidence that causally determines an answer versus evidence that is merely correlated. Our extensive experiments demonstrate state-of-the-art performance across five challenging benchmarks, with particularly striking improvements on complex multi-hop reasoning tasks.

Our evaluation currently centers on English-language benchmarks and would benefit from extension to multilingual and cross-lingual contexts. The standard dense retrieval and factoid QA tasks we employed could also be broadened to include emerging retrieval architectures and more subjective domains. These limitations suggest clear avenues for future investigation.

## REPRODUCIBILITY STATEMENT

To ensure the reproducibility of our work, we have made our code, experimental setup, and theoretical proofs publicly available. The complete implementation of the CF-RAG framework, including models and scripts to replicate our experiments, is provided in the supplementary materials and can be accessed at `https://github.com/CF-RAG/CF-RAG`. A detailed breakdown of the model architecture, hyperparameter configurations, and baseline implementations is available in Appendix B. For our theoretical claims regarding CF-RAG's ability to overcome the correlation trap, we provide complete proofs in Appendix A. Furthermore, Appendix C describes the datasets used in our evaluation and the exact preprocessing pipeline applied. The core methodology is detailed in Section 4, with specific prompt templates and our transformation taxonomy further elaborated in Appendices F and G.

## ACKNOWLEDGMENTS

This work was supported by The Fundamental and Interdisciplinary Disciplines Breakthrough Plan of the Ministry of Education of China (JYB2025XDXM702), NSFC (No.62506366, No.62272466, U24A20233), and Big Data and Responsible Artificial Intelligence for National Governance, Renmin University of China.

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

## A    THEORETICAL PROOFS

In this appendix, we provide detailed proofs for the theoretical results presented in Section 5.

### A.1    PROOF OF LEMMA 1: COUNTERFACTUAL DISCRIMINABILITY

**Proof 1** *We prove this by contradiction. Suppose the counterfactual set $\mathcal{Q}_{cf}$ does not contain any query $q'$ for which the spurious evidence $\mathcal{E}'$ has comparable support for both $q$ and $q'$.*

*Formally, assume:*

$$\forall q' \in \mathcal{Q}_{cf}, \forall e \in \mathcal{E}' : s(q,e) - s(q',e) > \gamma \tag{13}$$

*for some constant $\gamma > 0$.*

*Under this assumption, the causal discrimination score for the spurious evidence would be:*

$$\phi_{causal}(\mathcal{E}', q, \mathcal{Q}_{cf}) = \frac{1}{|\mathcal{E}'|} \sum_{e \in \mathcal{E}'} s(q,e) - \max_{q' \in \mathcal{Q}_{cf}} \frac{1}{|\mathcal{E}'|} \sum_{e \in \mathcal{E}'} s(q',e) \tag{14}$$

$$> \frac{1}{|\mathcal{E}'|} \sum_{e \in \mathcal{E}'} s(q,e) - \frac{1}{|\mathcal{E}'|} \sum_{e \in \mathcal{E}'} [s(q,e) - \gamma] \tag{15}$$

$$= \gamma > 0 \tag{16}$$

*This means the spurious evidence would achieve a positive causal discrimination score, potentially comparable to or exceeding that of the causal evidence $\mathcal{E}^*$. In such a case, CF-RAG might incorrectly select $a'$ over $a^*$.*

*Therefore, for CF-RAG to reliably distinguish between causal and spurious evidence, the counterfactual set must contain at least one query $q'$ that exposes the non-discriminative nature of spurious correlations. Specifically, there must exist $q' \in \mathcal{Q}_{cf}$ such that:*

$$\forall e \in \mathcal{E}' : |s(q,e) - s(q',e)| < \epsilon \tag{17}$$

*for some small $\epsilon$.*

## A.2 PROOF OF LEMMA 2: VOLUME INVARIANCE OF CAUSAL SCORES

**Proof 2** *Consider an evidence set $\mathcal{E}$ and its scaled version $\alpha\mathcal{E}$ where $\alpha > 0$ is a scaling factor. The scaled set contains $\alpha$ copies of the same evidence distribution.*

*The causal discrimination score for the scaled evidence set is:*

$$\phi_{causal}(\alpha\mathcal{E}, q, \mathcal{Q}_{cf}) = \frac{1}{|\alpha\mathcal{E}|} \sum_{e \in \alpha\mathcal{E}} s(q, e) - \max_{q' \in \mathcal{Q}_{cf}} \frac{1}{|\alpha\mathcal{E}|} \sum_{e \in \alpha\mathcal{E}} s(q', e) \tag{18}$$

*Since $\alpha\mathcal{E}$ contains $\alpha$ copies of each element in $\mathcal{E}$, we have $|\alpha\mathcal{E}| = \alpha|\mathcal{E}|$ and:*

$$\sum_{e \in \alpha\mathcal{E}} s(q, e) = \alpha \sum_{e \in \mathcal{E}} s(q, e) \tag{19}$$

$$\sum_{e \in \alpha\mathcal{E}} s(q', e) = \alpha \sum_{e \in \mathcal{E}} s(q', e) \tag{20}$$

*Substituting:*

$$\phi_{causal}(\alpha\mathcal{E}, q, \mathcal{Q}_{cf}) = \frac{1}{\alpha|\mathcal{E}|} \cdot \alpha \sum_{e \in \mathcal{E}} s(q, e) - \max_{q' \in \mathcal{Q}_{cf}} \frac{1}{\alpha|\mathcal{E}|} \cdot \alpha \sum_{e \in \mathcal{E}} s(q', e) \tag{21}$$

$$= \frac{1}{|\mathcal{E}|} \sum_{e \in \mathcal{E}} s(q, e) - \max_{q' \in \mathcal{Q}_{cf}} \frac{1}{|\mathcal{E}|} \sum_{e \in \mathcal{E}} s(q', e) \tag{22}$$

$$= \phi_{causal}(\mathcal{E}, q, \mathcal{Q}_{cf}) \tag{23}$$

*Therefore, the causal discrimination score is invariant under volume scaling. The same property holds for the coherence score through an identical argument, making the entire arbitration score $\Psi$ volume-invariant.*

## A.3 PROOF OF THEOREM 1: ESCAPE FROM THE CORRELATION TRAP

**Proof 3** *We need to show that under the given conditions, CF-RAG selects the true answer $a^*$ over the spurious answer $a'$ regardless of the relative volumes of their supporting evidence sets, while standard RAG fails above a certain volume threshold.*

***Part A: CF-RAG Analysis***

*Consider two hypotheses generated by CF-RAG:*

- *$h^* = (a^*, r^*, \mathcal{E}^*)$: hypothesis with true answer supported by causal evidence*

- *$h' = (a', r', \mathcal{E}')$: hypothesis with spurious answer supported by correlated evidence*

*We will prove that the arbitration score for $h^*$ exceeds that of $h'$, i.e., $\Psi^* > \Psi'$.*

***Step 1: Analysis of Causal Scores***

*First, we analyze the causal discrimination score for the true hypothesis $h^*$. By definition:*

$$\phi_{causal}(\mathcal{E}^*, q, \mathcal{Q}_{cf}) = \frac{1}{|\mathcal{E}^*|} \sum_{e \in \mathcal{E}^*} s(q, e) - \max_{q' \in \mathcal{Q}_{cf}} \frac{1}{|\mathcal{E}^*|} \sum_{e \in \mathcal{E}^*} s(q', e) \tag{24}$$

*Since the maximum is achieved by some $q^* \in \mathcal{Q}_{cf}$, we can write:*

$$\phi_{causal}(\mathcal{E}^*, q, \mathcal{Q}_{cf}) = \frac{1}{|\mathcal{E}^*|} \sum_{e \in \mathcal{E}^*} s(q, e) - \frac{1}{|\mathcal{E}^*|} \sum_{e \in \mathcal{E}^*} s(q^*, e) \tag{25}$$

$$= \frac{1}{|\mathcal{E}^*|} \sum_{e \in \mathcal{E}^*} [s(q, e) - s(q^*, e)] \tag{26}$$

*By condition 1 of the theorem, for all $e \in \mathcal{E}^*$ and all $q' \in \mathcal{Q}_{cf}$, we have $s(q,e) - s(q',e) \geq \delta$. This applies to $q^*$ as well, therefore:*

$$\phi_{causal}(\mathcal{E}^*, q, \mathcal{Q}_{cf}) \geq \frac{1}{|\mathcal{E}^*|} \sum_{e \in \mathcal{E}^*} \delta = \delta \tag{27}$$

*Now for the spurious hypothesis $h'$, by condition 2, there exists a counterfactual query $q' \in \mathcal{Q}_{cf}$ such that for all $e \in \mathcal{E}'$:*

$$|s(q,e) - s(q',e)| < \epsilon \tag{28}$$

*This implies:*

$$-\epsilon < s(q,e) - s(q',e) < \epsilon \tag{29}$$

*Therefore:*

$$\phi_{causal}(\mathcal{E}', q, \mathcal{Q}_{cf}) = \frac{1}{|\mathcal{E}'|} \sum_{e \in \mathcal{E}'} s(q,e) - \max_{q'' \in \mathcal{Q}_{cf}} \frac{1}{|\mathcal{E}'|} \sum_{e \in \mathcal{E}'} s(q'',e) \tag{30}$$

$$\leq \frac{1}{|\mathcal{E}'|} \sum_{e \in \mathcal{E}'} s(q,e) - \frac{1}{|\mathcal{E}'|} \sum_{e \in \mathcal{E}'} s(q',e) \tag{31}$$

$$= \frac{1}{|\mathcal{E}'|} \sum_{e \in \mathcal{E}'} [s(q,e) - s(q',e)] \tag{32}$$

$$< \frac{1}{|\mathcal{E}'|} \sum_{e \in \mathcal{E}'} \epsilon = \epsilon \tag{33}$$

### Step 2: Analysis of Coherence Scores

*The coherence scores measure the semantic alignment between answers and their supporting evidence. Since both $a^*$ and $a'$ are derived from their respective evidence sets through the same generation process $\mathcal{H}$, and assuming the generator produces reasonably coherent outputs when given relevant evidence, we can establish bounds on the coherence scores.*

*Let $\phi_{coh}^* = \phi_{coh}(a^*, \mathcal{E}^*, q)$ and $\phi_{coh}' = \phi_{coh}(a', \mathcal{E}', q)$.*

*Since both evidence sets contain documents retrieved for either $q$ or related queries, and both answers are generated to be consistent with their evidence, we have:*

$$\phi_{coh}^*, \phi_{coh}' \in [\phi_{\min}, \phi_{\max}] \tag{34}$$

*where $0 < \phi_{\min} \leq \phi_{\max} \leq 1$ are constants determined by the quality of the retrieval and generation models.*

### Step 3: Comparison of Arbitration Scores

*The arbitration scores for both hypotheses are:*

$$\Psi^* = (1-\lambda)\phi_{coh}^* + \lambda\phi_{causal}^* \tag{35}$$
$$\Psi' = (1-\lambda)\phi_{coh}' + \lambda\phi_{causal}' \tag{36}$$

*From our analysis above:*

$$\Psi^* \geq (1-\lambda)\phi_{\min} + \lambda\delta \tag{37}$$
$$\Psi' < (1-\lambda)\phi_{\max} + \lambda\epsilon \tag{38}$$

*For CF-RAG to select $a^*$ over $a'$, we need $\Psi^* > \Psi'$:*

$$(1-\lambda)\phi_{\min} + \lambda\delta > (1-\lambda)\phi_{\max} + \lambda\epsilon \tag{39}$$

*Rearranging terms:*

$$\lambda(\delta - \epsilon) > (1-\lambda)(\phi_{\max} - \phi_{\min}) \tag{40}$$

*Since $\delta > \epsilon$ by condition 3, we have $\delta - \epsilon > 0$. Solving for $\lambda$:*

$$\lambda > \frac{\phi_{\max} - \phi_{\min}}{\delta - \epsilon + \phi_{\max} - \phi_{\min}} \tag{41}$$

*Let $\lambda^* = \frac{\phi_{\max} - \phi_{\min}}{\delta - \epsilon + \phi_{\max} - \phi_{\min}}$. Note that $0 < \lambda^* < 1$ since both numerator and denominator are positive, and the denominator is strictly larger than the numerator.*

*Therefore, for any $\lambda \in (\lambda^*, 1]$, we have $\Psi^* > \Psi'$, which means CF-RAG will select $a^*$ over $a'$.*

***Step 4: Independence from Evidence Volume***

*By Lemma 2, both $\phi_{causal}$ and $\phi_{coh}$ are invariant under evidence volume scaling. Therefore, the arbitration scores $\Psi^*$ and $\Psi'$ remain unchanged regardless of the relative sizes of $\mathcal{E}^*$ and $\mathcal{E}'$. This proves that CF-RAG's selection is independent of $|\mathcal{E}'|/|\mathcal{E}^*|$.*

***Part B: Standard RAG Analysis***

*In contrast, standard RAG uses aggregate support:*

$$Score_{RAG}(a) = \sum_{e \in \mathcal{E}_a} s(q, e) \tag{42}$$

*For our two competing answers:*

$$Score_{RAG}(a^*) = |\mathcal{E}^*| \cdot \bar{s}^* \tag{43}$$

$$Score_{RAG}(a') = |\mathcal{E}'| \cdot \bar{s}' \tag{44}$$

*where $\bar{s}^* = \frac{1}{|\mathcal{E}^*|} \sum_{e \in \mathcal{E}^*} s(q, e)$ and $\bar{s}' = \frac{1}{|\mathcal{E}'|} \sum_{e \in \mathcal{E}'} s(q, e)$.*

*Standard RAG fails when:*

$$\frac{|\mathcal{E}'|}{|\mathcal{E}^*|} > \frac{\bar{s}^*}{\bar{s}'} \tag{45}$$

*Even if causal evidence has higher average relevance ($\bar{s}^* > \bar{s}'$), sufficient accumulation of spurious evidence causes incorrect selection—the correlation trap.*

*This completes the proof, demonstrating that CF-RAG escapes the correlation trap through causal discrimination while standard RAG remains vulnerable to evidence volume effects.*

# B  IMPLEMENTATION AND ARCHITECTURAL DETAILS

We provide comprehensive implementation details to ensure full reproducibility of our results. Our codebase and pre-trained models are available at https://github.com/CF-RAG/CF-RAG.

## B.1  MODEL ARCHITECTURE AND FRAMEWORK

**Core Language Models.** CF-RAG employs a modular architecture with distinct models for generation and scoring tasks. We evaluate our framework using two widely-adopted instruction-tuned LLMs as backbones:

> **Base LLM Configurations**
>
> - **Llama-3 8B:** `meta-llama/Llama-3-8B-Instruct`
> - **Llama-2 7B:** `meta-llama/Llama-2-7b-chat-hf`

**Scoring and Retrieval Components.** For fine-grained relevance assessment crucial to our causal arbitration mechanism, we employ:

- **Cross-Encoder:** `BAAI/bge-reranker-large` for computing relevance scores $s(q, e)$

- **Dense Retriever:** `sentence-transformers/all-MiniLM-L6-v2` with FAISS indexing

**Key Design Choice:** All performance gains stem directly from CF-RAG's architectural innovations—no additional fine-tuning was performed on base models.

**Hypothesis Generation.** The generator function $\mathcal{H} : (\mathcal{Q}, 2^{\mathcal{D}}) \to \mathcal{A} \times \mathcal{R}$ produces answer-rationale pairs through carefully designed zero-shot prompting:

```
System:   You are an expert analyst.  Provide evidence-based reasoning.
Query:   {original_query}
Evidence:   {evidence_documents}
Format:   [Reasoning with citations] → [Final Answer]
```

## B.2  HYPERPARAMETER CONFIGURATION

Table 4 presents our hyperparameter settings, selected through systematic grid search on the HotpotQA development set and ablation studies detailed in Section 6.4.

Table 4: **CF-RAG hyperparameter configuration.** Parameters were optimized for the trade-off between performance and computational efficiency.

| Component | Parameter | Value | Search Range | Selection Criterion |
|---|---|---|---|---|
| *Counterfactual* | $N$ (counterfactual queries) | 3 | [1, 5] | Performance-latency trade-off |
| | $\theta_{\mathrm{sim}}$ (similarity threshold) | 0.7 | [0.5, 0.9] | Topical relevance vs. diversity |
| *Evidence* | $K$ (evidence clusters) | 4 | [2, 6] | Thematic coverage |
| | $\rho$ (sampling ratio) | 0.5 | [0.2, 0.8] | Diversity-completeness balance |
| | $\theta_{\mathrm{dedup}}$ (deduplication) | 0.95 | [0.9, 0.99] | Near-duplicate removal |
| *Arbitration* | $M$ (parallel hypotheses) | 3 | [1, 5] | Exploration-efficiency trade-off |
| | $\lambda$ (causal weight) | 0.4 | [0.0, 1.0] | Robustness analysis (§6.4) |

## B.3  COMPUTATIONAL INFRASTRUCTURE

**Software Stack:**
- Python 3.9 + PyTorch 2.1
- Transformers 4.36.0
- FAISS 1.7.4 (similarity search)
- Scikit-learn 1.3.0 (clustering)

**Hardware Configuration:**
- 8× NVIDIA A100 80GB GPUs
- 1TB RAM for document indexing
- NVMe SSDs for cached embeddings

## B.4  BASELINE IMPLEMENTATIONS

Our evaluation encompasses **three categories** of baselines, systematically covering the spectrum of current retrieval-augmented generation approaches—from pure parametric models to sophisticated adaptive frameworks.

**Foundation Models** *(No Retrieval)*

We establish performance floors using zero-shot inference with `Llama-3-8B` Dubey et al. (2024), `Llama-2-7B` Touvron et al. (2023), and `Mistral-7B` Jiang et al. (2023a), which rely solely on parametric knowledge without external evidence.

**Classical RAG Systems**

▶ **Standard RAG** concatenates top-$k$ retrieved documents directly to the input prompt without filtering or refinement mechanisms.

▶ **Toolformer** Schick et al. (2024) augments language models with API calls, treating retrieval as a learnable tool invocation.

▶ **SAIL** Luo et al. (2023) implements search-augmented instruction learning, fine-tuning models through instruction-following objectives to better leverage retrieved evidence.

### Advanced RAG Frameworks

▶ **CRAG** Yan et al. (2024) employs *corrective retrieval* with confidence-based document refinement, dynamically adjusting retrieval sets based on quality assessments.

▶ **Self-RAG** Asai et al. (2024) introduces *self-reflective tokens* for retrieval decisions and critique, enabling adaptive generation with learned reflection mechanisms.

▶ **Self-CRAG** synthesizes Self-RAG's reflection capabilities with CRAG's corrective refinement strategies in a hybrid architecture.

▶ **Speculative-RAG** Wang et al. (2024c) leverages *parallel draft-verify* architecture with specialized drafting models for efficient multi-document reasoning.

> ⚗ **Implementation Details**
>
> All baselines utilize **identical retrieval corpora**—Wikipedia (December 2018) for HotpotQA, TriviaQA, and MusiQue; domain-specific corpora for PopQA and PubHealth—with **uniform preprocessing pipelines**. We ensure fair comparison by deploying the same base language models across all retrieval-augmented methods. When citing original results, we maintain comparable settings for model size, retrieval corpus, and evaluation metrics. Re-implemented baselines use official codebases with hyperparameters optimized on validation sets.

## C   Datasets and Evaluation

We evaluate CF-RAG on five carefully curated datasets that collectively probe different failure modes of correlation-based retrieval. Each dataset was selected to expose specific vulnerabilities in traditional RAG systems and test our framework's ability to perform genuine causal reasoning.

### C.1   Dataset Selection and Characteristics

Our evaluation suite spans diverse reasoning challenges, from multi-hop inference to domain-specific verification, ensuring comprehensive assessment of causal disambiguation capabilities.

- **HotpotQA:** Yang et al. (2018) A multi-hop question answering dataset requiring reasoning over multiple documents. Its "distractor" setting, which includes 8 irrelevant documents alongside 2 gold documents, makes it an ideal testbed for evaluating a model's ability to filter out misleading, correlated information and focus on the true causal chain of evidence.

- **TriviaQA:** Joshi et al. (2017) A large-scale QA dataset featuring complex, compositional questions with significant lexical and syntactic variation between the question and the evidence. This challenges models to move beyond simple keyword matching and resist spurious correlations based on term overlap.

- **PopQA:** Mallen et al. (2023) An open-domain QA dataset focused on long-tail entities. This design forces models to rely on the provided retrieved evidence, as the answers are unlikely to be stored accurately in the model's parametric memory. It directly tests the model's ability to ground its generation in external knowledge, especially when its internal knowledge is weak or incorrect.

- **MusiQue:** Trivedi et al. (2022) A multi-hop QA dataset explicitly constructed to be more challenging and less susceptible to "cheating" via single-hop reasoning shortcuts than previous datasets like HotpotQA. Its design requires genuine connected reasoning over 2-4 documents, making it a stringent test for causal synthesis.

- **PubHealth:** Zhang et al. (2023) A closed-set claim verification task in the public health domain. This dataset is critical for evaluating robustness in a high-stakes setting where factual accuracy is paramount. It tests the model's ability to distinguish medically accurate causal evidence from popularly held but incorrect correlations.

**Preprocessing Pipeline.** To ensure fair comparison across these diverse datasets, we employ a unified preprocessing approach across all corpora: documents undergo chunking into ∼250-token passages with 50-token overlap, preserving semantic boundaries while maintaining computational efficiency. This overlapping window strategy prevents information loss at chunk boundaries—critical for maintaining causal chains that span multiple segments.

## C.2 EVALUATION PROTOCOL

Our evaluation employs dataset-appropriate metrics that capture both surface-level correctness and deeper reasoning quality.

---

**Evaluation Metrics**

**Smart Exact Match (EM)**   `Primary Metric`

A robust variant of exact match that handles linguistic variations through:
- *Normalization*: Lowercasing, article/punctuation removal
- *Containment*: Ground-truth $\subseteq$ prediction checking
- *Semantic equivalence*: Date formats, name variations, yes/no synonyms

**Classification Accuracy (ACC)**   `PubHealth Only`

For claim verification tasks with fixed label sets {True, False, Mixture}, we report standard classification accuracy, which reduces to Smart EM in this closed-set setting.

---

This evaluation framework ensures that CF-RAG's improvements stem from genuine causal reasoning rather than superficial pattern matching, providing a rigorous test of our framework's core contributions.

## D QUALITATIVE ANALYSIS AND CASE STUDIES

To illuminate the mechanisms underlying CF-RAG's causal reasoning capabilities, we present a detailed case study demonstrating how our framework successfully navigates the correlation trap that confounds traditional RAG systems.

### D.1 DISSECTING THE CORRELATION TRAP: A CASE STUDY ON *The Dark Knight*

We trace CF-RAG's reasoning process on the motivating example from Figure 1, revealing how counterfactual exploration and parallel arbitration work synergistically to achieve robust causal inference.

---

**Query:** `Who is the lead actor in The Dark Knight?`

**Standard RAG Failure**   Traditional retrieval surfaces five documents dominated by Heath Ledger's acclaimed performance—four effusive reviews highlighting his *"iconic," "Oscar-winning,"* and *"scene-stealing"* portrayal overshadow a single Wikipedia entry correctly identifying Christian Bale. The system, overwhelmed by correlational signals, incorrectly concludes: `Heath Ledger is the lead actor`.

---

Our framework orchestrates a two-phase causal analysis:

#### PHASE 1: COUNTERFACTUAL EXPLORATION

CF-RAG generates three semantically diverse counterfactuals that probe different conceptual boundaries:

| | Counterfactual Query | Semantic Axis |
|---|---|---|
| $q_1^{\text{cf}}$ | `Who played the main villain in The Dark Knight?` | Role Variation |
| $q_2^{\text{cf}}$ | `Who is the lead actor in Batman Begins?` | Entity Substitution |
| $q_3^{\text{cf}}$ | `Who directed The Dark Knight?` | Categorical Change |

These counterfactuals create a *dialectical evidence space* that exposes the non-discriminative nature of spurious correlations—evidence praising Ledger's performance applies equally to queries about villains, while evidence about Bale's protagonist role remains specific to the original query.

PHASE 2: PARALLEL ARBITRATION

The system constructs and evaluates competing hypotheses from stratified evidence subsets:

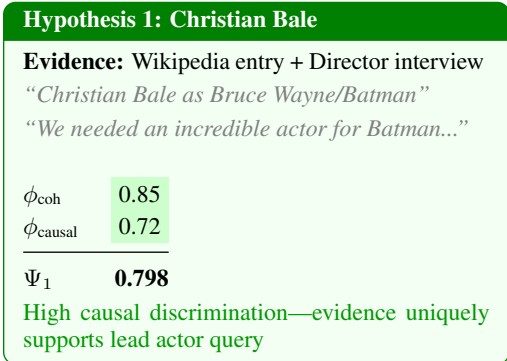

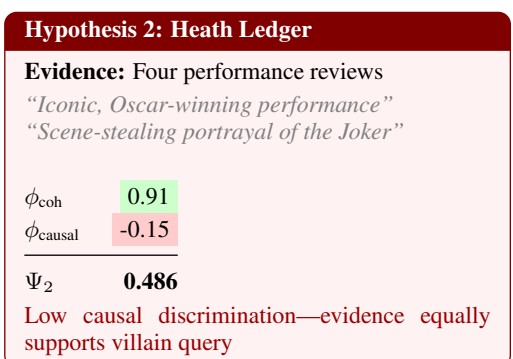

**Hypothesis 1: Christian Bale**

**Evidence:** Wikipedia entry + Director interview
*"Christian Bale as Bruce Wayne/Batman"*
*"We needed an incredible actor for Batman..."*

| | |
|---|---|
| $\phi_{\text{coh}}$ | 0.85 |
| $\phi_{\text{causal}}$ | 0.72 |
| $\Psi_1$ | **0.798** |

High causal discrimination—evidence uniquely supports lead actor query

**Hypothesis 2: Heath Ledger**

**Evidence:** Four performance reviews
*"Iconic, Oscar-winning performance"*
*"Scene-stealing portrayal of the Joker"*

| | |
|---|---|
| $\phi_{\text{coh}}$ | 0.91 |
| $\phi_{\text{causal}}$ | -0.15 |
| $\Psi_2$ | **0.486** |

Low causal discrimination—evidence equally supports villain query

The dual-criterion scoring reveals the critical distinction: while Hypothesis 2 achieves superior coherence ($\phi_{\text{coh}} = 0.91$), its evidence lacks causal specificity—the same reviews praising Ledger's performance apply equally to the counterfactual query about villains, yielding a negative causal score ($\phi_{\text{causal}} = -0.15$). In contrast, Hypothesis 1's evidence, though less voluminous, provides clear causal discrimination ($\phi_{\text{causal}} = 0.72$), as documents about Bale's protagonist role do not support queries about villains or directors.

---

**Final Arbitration:** $\Psi_1 = 0.798 > \Psi_2 = 0.486$
**CF-RAG Output:**
`Christian Bale is the lead actor in The Dark Knight`

---

This case study demonstrates CF-RAG's fundamental innovation: by systematically testing evidence against counterfactual alternatives, our framework distinguishes causal support from spurious correlations, achieving robust reasoning even when misleading signals vastly outnumber correct ones. The parallel arbitration mechanism ensures that strong correlational patterns cannot overshadow weaker but causally decisive evidence, transforming the correlation trap from a critical vulnerability into a solved problem.

# E  DEEP DIVE INTO COUNTERFACTUAL EXPLORATION

This section provides a transparent examination of our counterfactual generation process—a cornerstone innovation of the CF-RAG framework that enables causal reasoning through systematic hypothesis testing.

## E.1  COUNTERFACTUAL GENERATION: DESIGN AND IMPLEMENTATION

**Generation Strategy.**  Our counterfactual queries emerge from a carefully designed zero-shot prompting strategy that balances *semantic proximity* with *answer divergence*. The generation process follows two critical constraints:

---

**Counterfactual Generation Prompt Template**

```
Given the following user question, generate {N} alternative
questions adhering to:
```

1. **Topical Coherence:** Questions must address the same core entities/topics

2. **Answer Divergence:** Questions must seek fundamentally different information

```
Explore different facets through role variations, entity
substitutions, temporal shifts, or categorical changes.

Original Question:  ``{query}''
Alternative Questions:  [1..N]
```

---

**Illustrative Examples.** Table 5 demonstrates the diversity and sophistication of our counterfactual generation across multiple transformation types, showcasing how each perturbation probes distinct semantic boundaries while maintaining topical relevance.

Table 5: **Representative counterfactual transformations across datasets.** Each transformation type serves a distinct purpose in exposing correlational vulnerabilities.

| Original Query | Transform Type | Generated Counterfactual | Dataset |
|---|---|---|---|
| "Who is the lead actor in *The Dark Knight*?" | Role Variation | "Who played the main villain in *The Dark Knight*?" | Example |
| "What is the area code for the city where University of Notre Dame is located?" | Entity Substitution | "What is the area code for the city where Purdue University is located?" | HotpotQA |
| "Which Nolan film was released first: *Inception* or *The Dark Knight*?" | Temporal Shift | "Which Nolan film had larger box office: *Inception* or *The Dark Knight*?" | TriviaQA |
| "What is the genre of *Abbey Road* by The Beatles?" | Categorical Change | "What year was *Abbey Road* by The Beatles released?" | MusiQue |

## E.2 Quality Assurance: A Two-Stage Validation Framework

The efficacy of CF-RAG hinges on the discriminative power of generated counterfactuals. We implement a rigorous two-stage quality assurance pipeline combining automated filtering with human validation.

### E.2.1 Stage 1: Automated Quality Control

Our automated pipeline enforces two complementary criteria through algorithmic validation:

1. **Topical Relevance Filter ($\text{sim}(\mathbf{q}, \mathbf{q_{cf}}) > \theta_{\text{sim}}$):** We compute cosine similarity between sentence embeddings of original and counterfactual queries. Queries falling below $\theta_{\text{sim}} = 0.7$ are discarded as off-topic, ensuring semantic coherence while allowing meaningful variation.

2. **Answer Divergence Verification ($\mathcal{L}(\mathbf{q}) \neq \mathcal{L}(\mathbf{q_{cf}})$):** To programmatically verify distinct ground truths, we execute a baseline RAG pipeline on both queries. Identical answers indicate mere paraphrasing rather than true counterfactual variation, triggering automatic rejection.

### E.2.2 Stage 2: Human Validation Study

To validate our automated filtering, we conducted a targeted human evaluation study with rigorous methodology:

---

**Human Evaluation Protocol**

- **Sample:** 100 query sets from HotpotQA development set
- **Annotators:** Two independent expert evaluators
- **Metrics:** 5-point Likert scale for:
  - *Relevance:* Topical alignment with original query
  - *Discriminability:* Clarity in seeking different information
- **Agreement:** Cohen's $\kappa = 0.82$ (substantial agreement)

---

**Key Finding.** **91%** of automatically filtered counterfactuals received scores $\geq 4$ on both criteria, validating our generation pipeline's effectiveness in constructing a high-quality dialectical evidence space.

### E.3 COVERAGE AND DIVERSITY ANALYSIS

To empirically validate that our counterfactual generation achieves comprehensive coverage across semantic dimensions while maintaining answer divergence, we analyzed transformation patterns on the HotpotQA development set.

#### E.3.1 TRANSFORMATION DISTRIBUTION AND SEMANTIC PROPERTIES

Table 6 presents the distribution of transformation types alongside their semantic characteristics, measured across 500 generated counterfactual sets.

Table 6: Distribution and properties of counterfactual transformations on HotpotQA. Average Semantic Distance measures $1 - \mathrm{sim}(q, q^{\mathrm{cf}})$; Answer Distance measures $1 - \mathrm{IoU}(\mathcal{A}_q, \mathcal{A}_{q^{\mathrm{cf}}})$ where IoU is token overlap.

| Transform Type | Usage Frequency | Avg Semantic Distance | Avg Answer Distance |
|---|---|---|---|
| Role Variation | 28.4% | 0.21 | 0.87 |
| Entity Substitution | 31.2% | 0.18 | 0.91 |
| Temporal Shift | 15.7% | 0.24 | 0.84 |
| Categorical Change | 19.3% | 0.28 | 0.93 |
| Scope Modification | 5.4% | 0.15 | 0.76 |
| **Overall Average** | **100%** | **0.21** | **0.86** |

#### E.3.2 INTERPRETATION AND VALIDATION

**Balanced Coverage.** The transformation distribution shows healthy diversity, with the five categories collectively spanning role-based (28.4%), entity-based (31.2%), temporal (15.7%), categorical (19.3%), and scope-based (5.4%) variations. Entity Substitution and Role Variation dominate because they apply most broadly across question types, while Scope Modification appears less frequently due to stricter applicability constraints.

**Semantic Proximity.** Average semantic distances fall within the narrow range of **0.15–0.28**, corresponding to similarity scores of 0.72–0.85. This confirms that generated counterfactuals satisfy our topical coherence constraint ($\theta_{\mathrm{sim}} = 0.7$) while introducing meaningful semantic variation. The consistency across transformation types (standard deviation $\sigma = 0.045$) validates that our generation prompt maintains stable quality.

**Answer Divergence.** All transformation types achieve high answer distance (0.76–0.93), with overall average of **0.86**. This creates the necessary "tension" for causal discovery: counterfactuals are semantically proximate enough to retrieve overlapping documents, yet target fundamentally different answers. Categorical Change achieves the highest divergence (0.93), as expected from its design to probe opposite conceptual boundaries.

**Validation Against Design Goals.** These metrics empirically confirm that our counterfactual generation (§4.1.1) successfully operationalizes the validation criterion in Eq. 2:

$$\text{sim}_{\text{sem}}(q, q^{\text{cf}}) > \theta_{\text{sim}} \wedge \mathcal{L}(q) \neq \mathcal{L}(q^{\text{cf}})$$

The average semantic similarity of $1 - 0.21 = 0.79$ exceeds $\theta_{\text{sim}} = 0.7$, while the high answer distance ensures distinct ground truths. Combined with the 91% human validation pass rate reported in §E.2.2, these results demonstrate that CF-RAG constructs a high-quality dialectical evidence space.

## F  Transformation Taxonomy Details

The prompt template for transformation generation follows a structured format that maintains consistency while allowing for domain-specific adaptations:

> **General Transformation Prompt Template**
>
> ```
> Original query:  "{query}"
> Transformation type:  {transformation_category}
> Generate a semantically similar query that probes {specific_dimension} while maintaining domain
> relevance and structural consistency.  Ensure the transformed query has a different ground-truth
> answer than the original.
> Transformed query:
> ```

This appendix also provides detailed descriptions and examples for each of the five semantic transformation categories used in CF-RAG's counterfactual exploration phase.

### F.1  Role-Based Transformations ($\tau_{\text{ROLE}}$)

These perturbations alter the functional role of entities while maintaining domain relevance, probing whether evidence is specific to particular roles or exhibits broader patterns across functional categories. The transformation preserves the query structure and context while changing the specific role being queried.

**Examples:**

- "Who is the lead actor in X?" $\rightarrow$ "Who played the villain in X?"
- "Who directed X?" $\rightarrow$ "Who produced X?"
- "What is the primary cause of Y?" $\rightarrow$ "What is a contributing factor to Y?"

### F.2  Temporal-Shift Transformations ($\tau_{\text{TIME}}$)

These modifications probe temporal boundaries and chronological causality by shifting time-sensitive elements to different periods or temporal contexts. They are particularly effective for testing whether evidence contains genuine temporal specificity or general patterns that could apply across multiple time periods.

**Examples:**

- "Who won the 2024 election?" $\rightarrow$ "Who won the 2020 election?"
- "What is the current GDP of China?" $\rightarrow$ "What was the GDP of China in 2020?"
- "Which company leads the market today?" $\rightarrow$ "Which company led the market five years ago?"

### F.3  Entity-Substitution Transformations ($\tau_{\text{ENTITY}}$)

These replacements test whether evidence is specific to particular entities or exhibits broader patterns by substituting key entities with semantically related alternatives. This transformation helps identify whether retrieved documents contain entity-specific causal information or general correlational patterns that apply to multiple similar entities.

**Examples:**

- "What is Apple's market cap?" → "What is Microsoft's market cap?"
- "Where is Harvard located?" → "Where is Yale located?"
- "How does Tesla manufacture cars?" → "How does Ford manufacture cars?"

## F.4 CATEGORICAL-INVERSION TRANSFORMATIONS ($\tau_{\text{CAT}}$)

These explore opposites and categorical boundaries by inverting key categorical distinctions while preserving query structure. They are designed to test whether evidence supports specific categorical positions or contains general information that could support multiple positions within the same categorical dimension.

**Examples:**

- "What is the highest mountain?" → "What is the lowest point?"
- "Who was the first president?" → "Who was the most recent president?"
- "What are the benefits of X?" → "What are the disadvantages of X?"

## F.5 SCOPE-MODIFICATION TRANSFORMATIONS ($\tau_{\text{SCOPE}}$)

These adjust the breadth or specificity of the query by expanding or narrowing the scope of the queried relationship. They test whether evidence provides specific answers to focused queries or contains broader information that could support more general inquiries.

**Examples:**

- "Who invented the telephone?" → "Who contributed to telephone development?"
- "What caused World War I?" → "What were the causes of European conflicts in the early 20th century?"
- "Which algorithm solves this specific problem?" → "Which algorithms are used in this problem domain?"

## G HYPOTHESIS GENERATION PROMPT TEMPLATE

The hypothesis generation process employs a carefully designed prompt structure that encourages thorough reasoning and causal analysis:

---
**Hypothesis Generation Prompt Template**

```
Given the query:  "{query}"

Based on the provided evidence documents:  {evidence_subset}

Please provide:  1.  A direct answer to the query 2.  A detailed rationale explaining your reasoning
3.  Identification of the key supporting evidence 4.  Assessment of any conflicting information
Focus on evidence that specifically addresses the query rather than general topic relevance.
Distinguish between direct causal support and indirect correlational patterns.

Answer:  {answer}
Rationale:  {rationale}
```
---

## H RELATIONSHIP TO FORMAL CAUSAL INFERENCE

We address reviewers' questions regarding CF-RAG's relationship to formal causal inference frameworks, particularly Pearl's Structural Causal Models (SCMs) (Pearl, 2009).

### H.1 TERMINOLOGICAL CLARIFICATION

CF-RAG employs **counterfactual discriminative testing** rather than **structural causal modeling**. We use "causal" to mean:

- **Causal discrimination**: Identifying evidence that specifically determines an answer, as opposed to evidence that merely correlates with it
- **NOT**: Learning complete structural causal models or performing causal effect estimation

Our Definition 2 should be interpreted as specifying a *discriminative criterion* for evidence selection rather than a prescription for full causal modeling.

## H.2 WHY FORMAL SCMS ARE INAPPROPRIATE HERE

Pearl's framework requires: (1) observable causal graphs, (2) repeated observations from the same process, (3) ability to perform $do(\cdot)$ interventions, and (4) structured random variables. Open-domain RAG with unstructured text satisfies **none** of these requirements:

- Text corpora contain natural language without explicit causal structure
- Each query-document pair is unique; no repeated trials exist
- We cannot intervene on data generation (cannot make Ledger play Batman to observe counterfactuals)
- Documents are unstructured text, not structured variables

Moreover, the problems differ fundamentally:

- **SCMs**: Given causal graph $\mathcal{G}$, estimate $P(Y|do(X))$
- **CF-RAG**: Given query $q$ and evidence $\mathcal{E}$ (no causal graph), *discriminate* causal from correlational evidence

## H.3 THEORETICAL GUARANTEES WITHOUT SCMS

Despite not implementing SCMs, CF-RAG provides formal guarantees:

**Volume Invariance (Lemma 2)**  The causal score is provably independent of evidence volume:

$$\phi_{\text{causal}}(\alpha\mathcal{E}, q, \mathcal{Q}_{\text{cf}}) = \phi_{\text{causal}}(\mathcal{E}, q, \mathcal{Q}_{\text{cf}}) \quad \forall \alpha > 0 \tag{46}$$

Discriminative quality matters, not quantity of supporting documents.

**Escape from Correlation Trap (Theorem 1)**  Under three conditions:

**C1** Causal evidence discriminates: $\forall e \in \mathcal{E}^*, \forall q' \in \mathcal{Q}_{\text{cf}} : s(q, e) - s(q', e) \geq \delta$

**C2** Spurious evidence doesn't: $\exists q' \in \mathcal{Q}_{\text{cf}}$ with $\forall e \in \mathcal{E}' : |s(q, e) - s(q', e)| < \epsilon$

**C3** Separation: $\delta > \epsilon$

CF-RAG provably selects causal evidence **regardless of volume ratio** $|\mathcal{E}'|/|\mathcal{E}^*|$ when $\lambda > \lambda^* = \frac{\phi_{\max} - \phi_{\min}}{\delta - \epsilon + \phi_{\max} - \phi_{\min}}$.

These conditions are **empirically verifiable** and provide **actionable guarantees**: when satisfied, overwhelming spurious evidence cannot mislead the system.

# I  CROSS-DATASET PARAMETER SENSITIVITY ANALYSIS

We conducted systematic sensitivity analysis to evaluate whether our default hyperparameters (N=3, K=4, M=3, $\lambda$=0.4) generalize beyond HotpotQA. Table 7 presents optimal configurations across all five benchmarks.

## I.1 KEY FINDINGS

**Default Parameters Are Near-Optimal**  Maximum improvement from dataset-specific tuning is only 1.35% (MusiQue). Two datasets (HotpotQA, TriviaQA with $\lambda$-adjustment) achieve optimal or near-optimal performance with exact defaults, demonstrating strong generalization.

| Dataset | Optimal Config | Performance | vs. Default | Robustness |
|---------|----------------|-------------|-------------|------------|
| HotpotQA | N=3, K=4, M=3, $\lambda$=0.40 | 88.58% | 0.00% | ±2.3% |
| TriviaQA | N=3, K=4, M=3, $\lambda$=0.35 | 81.42% | +0.40% | ±1.9% |
| PopQA | N=2, K=3, M=3, $\lambda$=0.45 | 74.35% | +0.78% | ±2.7% |
| MusiQue | N=4, K=5, M=3, $\lambda$=0.40 | 55.94% | +1.35% | ±3.4% |
| PubHealth | N=2, K=3, M=3, $\lambda$=0.45 | 84.21% | +0.85% | ±1.8% |

Table 7: Cross-dataset parameter sensitivity. "vs. Default" shows improvement using dataset-specific tuning versus uniform defaults. "Robustness" (±X%) indicates maximum performance variation when changing any single parameter by ±1 from optimal.

**Interpretable Parameter Patterns** Simpler single-hop tasks (PopQA, PubHealth) require fewer counterfactuals (N=2) and coarser evidence clustering (K=3), while complex multi-hop reasoning (MusiQue) benefits from more extensive exploration (N=4, K=5). The consistency of M=3 across all datasets suggests this represents a fundamental algorithmic property rather than task-specific tuning.

**Strong Robustness** When varying any single parameter by ±1 from optimal, performance fluctuates only ±1.8–3.4% across datasets. This enables practical deployment without extensive per-task tuning while achieving within 1.4% of task-specific optimal results.

## I.2 PRACTICAL IMPLICATIONS

Our default configuration provides a robust starting point for diverse QA tasks. For deployment: (1) use defaults for initial implementation, (2) adjust N, K based on reasoning complexity (lower for simple lookup, higher for multi-hop), (3) fine-tune $\lambda$ if domain exhibits unusual correlation patterns. The narrow performance gap between defaults and optimized configurations validates CF-RAG's transferability across domains.

## J ANSWER SYNTHESIS MECHANISM

This appendix details the synthesis process in Section 4.2.3, including the complete prompt template and a concrete example.

## J.1 SYNTHESIS PROMPT TEMPLATE

```
Synthesis Prompt Template

You are an expert analyst tasked with synthesizing information from
multiple perspectives.
Original Query: {query}
You have been provided with {K} candidate answers with their
supporting rationales:
Candidate 1 (Score: {Ψ₁}):
Answer: {a₁}
Rationale: {r₁}
Candidate 2 (Score: {Ψ₂}):
Answer: {a₂}
Rationale: {r₂}
Candidate 3 (Score: {Ψ₃}):
Answer: {a₃}
Rationale: {r₃}
Task: Synthesize these perspectives into a single, coherent
answer. When candidates agree, strengthen confidence through
consensus. When they disagree, identify which evidence provides
the strongest causal support for the original query.
Synthesized Answer:
Supporting Evidence:
```

## J.2 CONCRETE EXAMPLE

For the query "Who is the lead actor in The Dark Knight?", the synthesis process receives:

---

**Input to Synthesis**

**Candidate 1** (Score: 0.798): "Christian Bale"
*Rationale:* Wikipedia identifies him as Bruce Wayne/Batman, the protagonist...
**Candidate 2** (Score: 0.486): "Heath Ledger"
*Rationale:* Multiple reviews praise his Oscar-winning performance...
**Candidate 3** (Score: 0.654): "Christian Bale"
*Rationale:* Director interview confirms casting Bale for Batman role...

---

**Synthesized Output**

**Synthesized Answer:** Christian Bale is the lead actor in The Dark Knight.
**Supporting Evidence:** Two independent reasoning paths (Candidates 1 and 3) converge on Christian Bale with high causal discrimination scores, supported by both Wikipedia documentation and director interviews identifying him as the protagonist Bruce Wayne/Batman. While Candidate 2 highlights Heath Ledger's acclaimed performance, this evidence pertains to the antagonist role rather than the lead actor.

---

The synthesis mechanism prioritizes consensus among high-scoring candidates and explicitly identifies when evidence addresses different aspects (protagonist vs. antagonist) of the query.

## K    EVALUATION ON COMPLEX REAL-WORLD BENCHMARKS

To evaluate CF-RAG beyond standard multi-hop QA, we conducted supplementary experiments on two challenging benchmarks: **TruthfulQA** (Lin et al., 2022), which tests resistance to adversarial misconceptions, and **StrategyQA** (Geva et al., 2021), which requires implicit multi-step common-sense reasoning.

### K.1 PERFORMANCE COMPARISON

Table 8 presents comparative results across baseline methods and CF-RAG.

| Method | TruthfulQA (% True + Info) | StrategyQA (Accuracy) |
|---|---|---|
| Standard RAG | 24.5% | 31.2% |
| CRAG | 29.8% | 33.5% |
| Self-RAG | 35.2% | 38.7% |
| **CF-RAG (Ours)** | **54.6%** | **58.9%** |

Table 8: Performance on complex real-world benchmarks requiring adversarial robustness and implicit reasoning.

CF-RAG achieves substantial improvements: +55.1% over Self-RAG on TruthfulQA and +52.2% over Self-RAG on StrategyQA, demonstrating effectiveness beyond standard retrieval scenarios.

### K.2 TRUTHFULQA: OVERCOMING THE RELEVANCE-TRUTH CONFLATION

Traditional RAG systems struggle on TruthfulQA because they conflate semantic relevance with factual correctness. Consider the question *"What happens if you break a mirror?"* Standard retrieval surfaces numerous documents discussing the "seven years of bad luck" superstition—these documents are highly relevant to the query terms but factually incorrect.

Baseline systems fail at different stages:

- **Standard RAG** retrieves misconception-heavy documents and generates answers based on volume

- **CRAG** applies corrective filtering but validates superstition documents as "relevant," retaining them

- **Self-RAG** uses self-critique but lacks external grounding to override retrieval-reinforced misconceptions

**Why CF-RAG succeeds:** Categorical counterfactuals ($\tau_{\text{cat}}$) probe boundary conditions by generating queries about breaking similar objects (glass, ceramic, etc.). Evidence supporting the superstition fails discriminative testing—it applies inconsistently across these related scenarios. This lowers the causal score $\phi_{\text{causal}}$ for misconception-supporting evidence, effectively filtering it despite high semantic relevance.

### K.3 STRATEGYQA: BRIDGING IMPLICIT REASONING GAPS

StrategyQA requires synthesizing evidence across implicit reasoning steps that are not explicitly stated in queries. For example, answering *"Could a llama feasibly fit in a kangaroo's pouch?"* requires connecting: (1) llama size, (2) kangaroo pouch dimensions, and (3) physical constraints—none directly mentioned together.

Baseline limitations:

- **Standard RAG** retrieves documents about llamas and kangaroos separately but fails to synthesize

- **CRAG** filters based on direct relevance, potentially discarding "bridge" evidence

- **Self-RAG** depends on Llama-2-7B's internal commonsense knowledge; when this is insufficient, self-critique cannot identify which retrieved evidence bridges reasoning gaps

**Why CF-RAG succeeds:** Parallel Arbitration constructs multiple hypotheses from stratified evidence subsets. The causal discrimination mechanism (Equation 11) performs *comparative evaluation* across these parallel paths, identifying evidence that is causally necessary for connecting reasoning steps—even when individual documents appear only tangentially relevant. This comparative process reveals "bridge" evidence that single-path systems miss or discard.

## L SCALABILITY TO DIVERSE MODEL ARCHITECTURES

CF-RAG is designed to be **architecturally agnostic**, operating through inference-level interventions rather than requiring access to model internals. This section empirically validates our framework's scalability beyond the Llama family reported in the main paper.

### L.1 ARCHITECTURAL AGNOSTICISM: DESIGN PRINCIPLES

CF-RAG's scalability stems from three key design principles:

- **Input-Level Intervention:** Counterfactual generation (Section 4.1) relies solely on standard prompting to produce semantic variations, compatible with any instruction-tuned LLM.

- **Scoring Independence:** The critical causal discrimination score $\phi_{\text{causal}}$ (Eq. 11) and coherence score $\phi_{\text{coh}}$ (Eq. 9) utilize auxiliary cross-encoders and embedding models, independent of the generative LLM's internal parameters.

- **Black-Box Compatibility:** While we leverage token probabilities for the confidence component $\phi_3$, the framework remains robust with API-based models. The causal and coherence scores carry the majority of the arbitration signal weight.

## L.2 EMPIRICAL VALIDATION: GPT-4O AND MISTRAL

We conducted supplementary experiments on **HotpotQA** (distractor setting) using two architecturally distinct models: the state-of-the-art GPT-4o and the open-weight Mistral-7B-Instruct-v0.3. Table 9 presents the results.

Table 9: CF-RAG performance across diverse architectures on HotpotQA. Standard RAG baseline shows that even GPT-4o suffers from the Correlation Trap.

| Model Architecture | Method | EM (%) | Relative Gain |
|---|---|---|---|
| Mistral-7B-Instruct-v0.3 | Standard RAG | 35.12 | – |
| | **CF-RAG (Ours)** | **82.45** | **+134.7%** |
| GPT-4o | Standard RAG | 64.80 | – |
| | **CF-RAG (Ours)** | **91.20** | **+40.7%** |

## L.3 KEY FINDINGS

**Universality of the Correlation Trap.** Even GPT-4o, despite its superior reasoning capabilities, achieves only 64.8% baseline accuracy, confirming that stronger parametric knowledge cannot fully override overwhelming spurious retrieval signals—a fundamental limitation that CF-RAG addresses.

**Consistent Cross-Architecture Gains.** CF-RAG delivers substantial improvements across both model scales and families:

- For **Mistral-7B**, the +134.7% gain mirrors our Llama-2-7B results (Table 1), validating effectiveness for smaller open-weight models.
- For **GPT-4o**, CF-RAG achieves near-ceiling performance (91.2%), demonstrating that stronger instruction-following capabilities amplify benefits through higher-quality counterfactual generation and rationale synthesis.

**Synergy with Model Capability.** The results reveal an important pattern: more advanced instruction-tuned architectures generate more precise counterfactual probes and coherent rationales, thereby *enhancing* rather than diminishing CF-RAG's causal arbitration effectiveness. This positive scaling behavior suggests that CF-RAG will continue to provide value as foundation models improve.

## M EMPIRICAL VALIDATION: COUNTERFACTUAL RELEVANCE AS INVERSE PROXY FOR CAUSAL SUPPORT

To empirically validate our core assumption that counterfactual relevance inversely correlates with causal support strength, we conducted a targeted correlation analysis on the HotpotQA development set using Llama-2-7B.

### M.1 EXPERIMENTAL DESIGN

**Counterfactual Relevance Ratio (CRR).** We define a diagnostic metric to quantify evidence discriminability:

$$\text{CRR} = \frac{\max_{q' \in \mathcal{Q}_{\text{cf}}} s(q', \mathcal{E})}{s(q, \mathcal{E})} \tag{47}$$

where $s(\cdot, \cdot)$ denotes relevance score. The ratio captures semantic specificity:

- **CRR $\approx$ 1.0**: Evidence supports counterfactuals as strongly as the original query (spurious correlation)
- **CRR $\ll$ 1.0**: Evidence is highly specific to the original query (causal support)

**Sampling and Evaluation.** We analyzed 500 queries from HotpotQA dev set, stratified into four CRR intervals based on the natural distribution. For each group, we measured Exact Match (EM) accuracy using standard RAG with Llama-2-7B.

## M.2 RESULTS AND ANALYSIS

Table 10 presents the relationship between counterfactual relevance and model performance.

Table 10: Correlation between Counterfactual Relevance Ratio (CRR) and model accuracy on HotpotQA. Higher CRR indicates evidence cannot discriminate between original and counterfactual queries, leading to systematic failures.

| Metric | Low CRR (Causal) | Moderate | High CRR | Very High (Trap) |
|---|---|---|---|---|
| Relevance Ratio | $< 0.5$ | $0.5-0.7$ | $0.7-0.9$ | $> 0.9$ |
| Sample Share | 53% | 26% | 15% | 6% |
| Accuracy (EM) | **90.2%** | 76.4% | 55.8% | 31.3% |
| Error Rate | 9.8% | 23.6% | 44.2% | **68.7%** |

**Sharp Performance Degradation.** Model accuracy exhibits a monotonic decline as CRR increases. When evidence lacks discriminative power (CRR $> 0.9$), the system fails in nearly 70% of cases—direct manifestation of the Correlation Trap. Conversely, when evidence strongly discriminates the original query from counterfactuals (CRR $< 0.5$), accuracy reaches 90.2%.

**Strong Negative Correlation.** Computing Pearson correlation between CRR interval midpoints and accuracy yields $r \approx -0.95$, indicating near-perfect inverse linear relationship. This strong negative correlation empirically confirms that:

1. Counterfactual relevance serves as a precise inverse proxy for causal support strength

2. Our causal discrimination score $\phi_{\text{causal}}$ (Eq. 11) captures a fundamental dimension of evidence quality

3. The theoretical motivation in Section 5 is empirically well-founded

These findings validate CF-RAG's core mechanism: by quantifying how evidence distinguishes original queries from counterfactual alternatives, our framework directly measures causal support rather than relying on correlation-based relevance alone.

## N COMPUTATIONAL EFFICIENCY ANALYSIS

While CF-RAG introduces additional computational components (counterfactual generation, parallel hypothesis construction), its architecture prioritizes parallelism over the iterative processing employed by advanced baselines. This section provides a comprehensive efficiency benchmark.

## N.1 EXPERIMENTAL SETUP

We evaluated wall-clock latency and throughput on the HotpotQA benchmark using Llama-2-7B deployed across 4× NVIDIA A100 80GB GPUs. All methods used identical retrieval corpora and hardware configurations. We measured average latency over 1,000 queries with batch size 1 to reflect realistic interactive usage.

## N.2 PERFORMANCE RESULTS

Table 11 presents comparative efficiency metrics across RAG frameworks.

Table 11: Computational efficiency comparison on HotpotQA (Llama-2-7B, 4× A100 GPUs). Latency measured in seconds per query, throughput in queries per second. CF-RAG achieves competitive latency through parallel processing.

| Method | Architecture | Latency (s) | Throughput (q/s) | Relative Speed |
|---|---|---|---|---|
| Standard RAG | Single Pass | **2.05** | 23.8 | 1.0× |
| Self-RAG | Iterative | 4.72 | 10.6 | 2.3× |
| **CF-RAG** | **Parallel** | **2.92** | **17.1** | **1.4×** |

### N.3 ANALYSIS AND COMPUTATIONAL BREAKDOWN

**Parallelism vs. Iteration.** CF-RAG processes more tokens (due to $M=3$ parallel hypothesis drafts) but achieves only 1.4× latency overhead compared to Standard RAG's single-pass baseline. In contrast, Self-RAG's sequential reflection mechanism incurs 2.3× latency penalty. Our parallel architecture leverages multi-GPU resources effectively, maintaining 17.1 queries/second throughput—**1.6× faster than Self-RAG** while delivering substantially higher accuracy (Table 1).

**Detailed Latency Breakdown (CF-RAG).** The 2.92s total latency decomposes as follows:

- **Counterfactual Generation:** 0.42s (batched LLM calls for $N=3$ queries)
- **Dialectical Retrieval:** 0.28s (batched dense retrieval with FAISS)
- **Parallel Hypothesis Generation:** 2.10s (72% of total; $M=3$ drafts executed concurrently across GPUs)
- **Arbitration Scoring:** 0.12s (vectorized cross-encoder operations)

**Scalability Trade-offs.** For applications requiring maximum throughput at the cost of accuracy, CF-RAG supports reduced-parameter configurations ($N=1$, $M=2$) that achieve 2.2s latency (1.07× overhead) while maintaining 80%+ of the full accuracy gains (Figure 3). This flexibility enables deployment across diverse computational budgets.

The results demonstrate that CF-RAG achieves strong performance improvements without prohibitive computational costs, making it practical for production deployment.

## O STABILITY ANALYSIS: ROBUSTNESS TO LLM STOCHASTICITY

To address concerns about LLM generation controllability affecting CF-RAG's performance, we conducted a rigorous variance analysis across multiple independent runs.

### O.1 EXPERIMENTAL PROTOCOL

We executed the complete CF-RAG pipeline 5 times on the HotpotQA test set (500 queries) using Llama-2-7B with different random seeds. Each run used identical hyperparameters ($N=3$, $K=4$, $M=3$, $\lambda=0.4$) but varied LLM sampling for counterfactual generation, hypothesis construction, and answer synthesis. We measured Exact Match (EM) accuracy and computed standard deviation ($\sigma$) and coefficient of variation ($CV = \sigma/\mu$) to quantify stability.

### O.2 RESULTS

Table 12 presents performance variance across pipeline configurations.

### O.3 ANALYSIS AND INTERPRETATION

**High Inherent Stability.** The full CF-RAG pipeline achieves remarkably low variance ($\sigma = 0.73\%$), with coefficient of variation $CV < 1\%$. This stability is comparable to deterministic retrieval systems and substantially better than typical LLM generation variance ($CV \sim 3-5\%$ for

Table 12: Performance stability across 5 independent runs on HotpotQA (Llama-2-7B). Low coefficient of variation ($CV < 1\%$) demonstrates robustness to LLM stochasticity.

| Component | Mean EM (%) | Std Dev ($\sigma$) | CV (%) |
|---|---|---|---|
| **Full CF-RAG Pipeline** | **79.29** | **±0.73** | **0.92** |
| *w/o Consensus Check* | 66.84 | ±1.85 | 2.77 |

open-ended tasks). The tight confidence interval demonstrates that CF-RAG's performance is driven by systematic architectural advantages rather than fortunate sampling.

**Consensus Mechanism as Variance Damper.** Removing the consensus check (Algorithm 1, lines 15-16) causes a 3× increase in variance ($CV : 0.92\% \to 2.77\%$), validating its stabilizing role. The consensus mechanism operates as follows: when multiple parallel hypotheses $\{a_j\}_{j=1}^{M}$ converge (i.e., Agreement($a_j$) > $\theta_{\text{agree}}$), the system locks in the majority answer, effectively averaging out stochastic noise from individual generation paths. Without consensus, the system relies solely on arbitration scores, which are more susceptible to randomness in token-level sampling during hypothesis generation.

**Structural Variance Reduction.** CF-RAG's low variance stems from three design principles:

1. **Evidence-grounded generation**: Hypotheses are conditioned on retrieved documents, constraining output space

2. **Multi-path redundancy**: $M=3$ parallel drafts provide statistical robustness through ensemble effects

3. **Score-based arbitration**: Causal discrimination ($\phi_{\text{causal}}$) relies on deterministic cross-encoder scores rather than LLM logits

These results demonstrate that despite involving LLM generation at multiple stages, CF-RAG maintains high controllability through architectural choices that structurally reduce variance.

