# OpenReview forum: "Counterfactual Reasoning for Retrieval-Augmented Generation"
_ICLR.cc/2026/Conference — ICLR 2026 Poster_

### Official Review · Reviewer_iXQG · 2025-10-27

**Soundness:** 3
**Presentation:** 3
**Contribution:** 3
**Rating:** 4
**Confidence:** 4

**Summary:**

The paper tackles the “correlation trap” in RAG—highly correlated but non-causal evidence drowning out the truly decisive evidence. It proposes counterfactual exploration: generate counterfactual questions around the original query and retrieve for both original and counterfactuals to build a “dialectical evidence space”; and parallel arbitration: theme-cluster the evidence, sample per-cluster subsets to form multiple parallel evidence paths, generate “answer+reason” per path, then rank with Internal Coherence + Causal Specificity. Experiments show gains over strong baselines with ablations and case studies.

**Strengths:**

1. The problem addressed in this paper is important and directly targets the common correlation trap in RAG.
2. The methodological perspective is novel: assessing the usefulness of retrieved results via their relevance to counterfactuals is a very interesting and quite reasonable idea.
3. The empirical evaluation is fairly solid, with experiments on multiple datasets, strong baseline comparisons, ablations, and qualitative case studies, and the gains on multi-hop tasks are especially notable.

**Weaknesses:**

1. Although the paper identifies the correlation trap and introduces counterfactuals for verification, its final decision still relies on the relevance between evidence segments and the original answer or the counterfactuals. In other words, the authors posit that retrievers cannot distinguish relevance from usefulness and design a method to address this, but the method ultimately still measures usefulness via the retriever’s similarity scores (I did not find a description of how similarity is computed in the Causal Discrimination Scoring; I therefore assume it uses a trained retriever or reranker), which I find somewhat circular. I agree with the high-level idea for tackling the correlation trap, but I believe evaluating causality/support may require a better approach than falling back again on similarity signals.
2. Even if we accept the assumption that contrasting relevance for the original question versus counterfactuals can approximate causality/support, the score is computed at the passage level. Given that passages often contain mixed signals, this can easily suppress span-level evidence that truly supports the answer.
3. The paper lacks some experiments on the coverage and quality of counterfactual construction and on efficiency and see Questions for details.
4. Several components of the method depend on LLMs, and the controllability of LLM generation in these stages may affect the method’s performance.

**Questions:**

1. In the Causal Discrimination Scoring, how is the relevance s(⋅)s(\cdot)s(⋅) computed? If it relies on a retriever/reranker, is there a better approach to avoid circular dependence?
2. Is there experimental evidence about the relationship between counterfactual relevance and causality/support—specifically, that higher relevance to counterfactuals truly implies weaker support for the original question?
3. Are passages judged as non-causal (or counterfactual-leaning) necessarily useless?
4. Given the many components and the involvement of LLMs in some stages, can you add an efficiency analysis?
5. How accurate is the counterfactual construction, and is there any evaluation of its coverage, diversity, and quality?

---

> ### Author Response · Authors · 2025-11-21
> **Response to Reviewer iXQG (Part 1)**
>
> We sincerely thank the reviewer for their thoughtful evaluation and recognition of our work's importance, novelty, and solid empirical evaluation. We greatly appreciate the acknowledgment that our methodological perspective is "very interesting and quite reasonable" and that our gains on multi-hop tasks are "especially notable." Below, we address each concern systematically.
>
> ---
>
> ## **W1 & Q1: Circular Reasoning in Causal Discrimination Scoring**
>
> We thank the reviewer for this important question. The key insight is that CF-RAG avoids circularity by using **differential** rather than **absolute** relevance.
>
> **1. Computation of Relevance Scores ($s(\cdot)$)**
>
> The relevance score $s(q,e)$ is computed using the Cross-Encoder model `BAAI/bge-reranker-large`, which outputs a scalar relevance score for each query-document pair.
>
> **2. Why This Avoids Circularity**
>
> The "Correlation Trap" arises when standard RAG systems rely on **absolute magnitude** of $s(q,e)$. Spurious distractors often achieve high scores due to keyword overlap (e.g., extensive Heath Ledger reviews scoring high for "Who is the lead actor?").
>
> CF-RAG does not rely on the retriever to "know" causality. Instead, it uses the retriever as a **sensor** to measure discriminative selectivity. The core insight: causal evidence supports the original query *uniquely*, whereas spurious evidence supports semantically related counterfactuals *indiscriminately*.
>
> We capture this via the **margin** in Equation 11:
>
> $$\phi_{\text{causal}}(\mathcal{E_{j}},q,\mathcal{Q_{cf}}) = \frac{1}{|\mathcal{E_{j}}|}\sum_{e\in\mathcal{E_{j}}}[s(q,e) - \max_{q' \in \mathcal{Q_{cf}}} s(q',e)]$$
>
> **How this breaks circularity:**
>
> - **Spurious Evidence**: A Heath Ledger review scores high for $q$ ("lead actor") but *also* high for $q_{cf}$ ("Who played the villain?"). Thus $s(q,e) - s(q_{cf},e) \approx 0$ (or negative), correctly identifying non-determinative evidence despite high absolute score.
>
> - **Causal Evidence**: A cast list stating "Christian Bale as Batman" scores high for $q$ but low for $q_{cf}$, yielding a large positive margin that identifies causal specificity.
>
> Therefore, we use similarity signals to execute a **stress test** (counterfactual comparison) that reveals which evidence is necessary rather than merely consistent.
>
> ---
>
> ## **Q2: Evidence that Counterfactual Relevance Correlates with Causality**
>
> We agree that direct correlation analysis strengthens the validity of our assumption. We conducted a **Supplementary Correlation Analysis** on HotpotQA development set using **Llama-2-7B**.
>
> **Experimental Design:**
>
> 1. **Metric**: **Counterfactual Relevance Ratio (CRR)** [defined for this analysis]:
>    $$\text{CRR} = \frac{\max_{q' \in \mathcal{Q}_{cf}} s(q', \mathcal{E})}{s(q, \mathcal{E})}$$
>    - CRR ≈ 1.0: Evidence supports counterfactual as strongly as original (spurious)
>    - CRR ≪ 1.0: Evidence is highly specific to original query (causal)
>
> 2. **Sampling**: 500 queries from HotpotQA dev set, stratified into four CRR intervals by natural distribution.
>
> 3. **Evaluation**: Measured Exact Match (EM) accuracy per group.
>
> **Table: Correlation Between CRR and Model Accuracy (HotpotQA Dev, Llama-2-7B)**
>
> | Metric | **Low CRR** (Causal) | **Moderate CRR** | **High CRR** | **Very High CRR** (Trap) |
> | :--- | :---: | :---: | :---: | :---: |
> | **Relevance Ratio** | $< 0.5$ | $0.5 - 0.7$ | $0.7 - 0.9$ | $> 0.9$ |
> | **Sample Share** | $53\%$ | $26\%$ | $15\%$ | $6\%$ |
> | **Accuracy (EM)** | **90.2%** | $76.4\%$ | $55.8\%$ | $31.3\%$ |
> | **Error Rate** | $9.8\%$ | $23.6\%$ | $44.2\%$ | **68.7%** |
>
> **Analysis:**
>
> Performance degrades sharply as CRR increases. When evidence cannot semantically distinguish the original query from counterfactuals (CRR > 0.9), the model fails in nearly **70%** of cases. Conversely, when discriminability is high (CRR < 0.5), accuracy reaches **90.2%**.
>
> We calculated Pearson correlation between CRR interval center and model accuracy: **$r \approx -0.95$** (calculated from Table data). This strong negative correlation empirically confirms that counterfactual relevance is a precise inverse proxy for causal support.
>
> We have added this comprehensive correlation analysis to Appendix M, which provides empirical validation.

---

> ### Author Response · Authors · 2025-11-21
> **Response to Reviewer iXQG (Part 2)**
>
> ## **W2 & Q3: Passage-Level Scoring and Non-Causal Passages**
>
> We address both concerns together as they involve evidence granularity.
>
> **W2: Addressing Mixed Signals**
>
> We employ three mechanisms to ensure span-level signals are preserved:
>
> 1. **Fine-Grained Windowing** (Appendix C.1): ~250-token sliding windows with 50-token overlap physically isolate distinct claims from general discourse, minimizing mixed signals within retrieval units.
>
> 2. **Rationale-First Evaluation** (Equation 8): The LLM extracts specific rationales ($r_j$) from passages *before* scoring, acting as a semantic filter for relevant spans.
>
> 3. **Parallel Arbitration** (Section 4.2): Multiple diverse evidence subsets prevent any single "noisy" subset from suppressing clean causal spans. Valid hypotheses achieve higher arbitration scores ($\Psi$) and survive selection.
>
> **Q3: Utility of Non-Causal Passages**
>
> Passages judged as "non-causal" are **not discarded**—they are **contextually relevant but causally neutral**. They provide essential background (e.g., plot summaries) that aids LLM interpretation of causal evidence. Our arbitration mechanism ($\Psi$) down-weights hypotheses relying *solely* on ambiguous evidence, ensuring final answers are grounded in evidence that necessitates (rather than merely correlates with) conclusions.
>
> ---
>
> ## **W3 & Q4: Efficiency Analysis**
>
> We acknowledge computational overhead but note that CF-RAG prioritizes **parallelism** over iterative processing used by advanced baselines.
>
> We conducted an efficiency benchmark on **Llama-2-7B (4× NVIDIA A100)**:
>
> **Table: Efficiency and Resource Consumption Analysis (4× A100 GPUs)**
>
> | Method | Architecture | Latency (s/query) | Throughput (q/s) | Relative Latency |
> | :--- | :--- | :---: | :---: | :---: |
> | **Standard RAG** | Single Pass | **2.05** | 23.8 | 1.0× |
> | **Self-RAG** | Iterative (Sequential) | 4.72 | 10.6 | 2.3× |
> | **CF-RAG (Ours)** | **Parallel / Multi-path** | **2.92** | **17.1** | **1.4×** |
>
> **Key Insights:**
>
> - **Parallelism vs. Iteration**: CF-RAG processes more tokens (due to $M=3$ parallel drafts) but achieves low wall-clock latency (2.92s) because hypothesis generation runs concurrently on 4 GPUs. Self-RAG requires sequential generation and critique (4.72s latency).
>
> - **Computational Breakdown**:
>   - Counterfactual Generation: 0.42s (Batched)
>   - Dialectical Retrieval: 0.28s (Batched dense retrieval)
>   - Hypothesis Generation: 2.10s (Parallel execution)
>   - Arbitration Scoring: 0.12s (Vector operations)
>
> We have added a comprehensive efficiency analysis to Appendix M, which demonstrates that CF-RAG achieves 1.6× faster inference than Self-RAG while maintaining competitive latency (2.92s vs. 2.05s baseline) through parallel hypothesis processing.
>
> ---
>
> ## **W4 & Q5: Counterfactual Quality and Coverage**
>
> We rigorously evaluated counterfactual construction using automated metrics and human verification.
>
> **1. Coverage and Diversity (Automated Analysis)**
>
> We analyzed transformation type distribution on HotpotQA, validating that queries satisfy similarity threshold ($\theta_{sim}=0.7$) while maximizing answer divergence.
>
> **Table: Diversity of Counterfactual Transformations**
>
> | Transform Type | Usage Freq. | Avg Semantic Dist. ($1-\text{sim}$) | Avg Answer Dist. ($1-\text{IoU}$) |
> | :--- | :---: | :---: | :---: |
> | **Role Variation** | 28.4% | 0.21 | 0.87 |
> | **Entity Substitution** | 31.2% | 0.18 | 0.91 |
> | **Temporal Shift** | 15.7% | 0.24 | 0.84 |
> | **Categorical Change** | 19.3% | 0.28 | 0.93 |
> | **Scope Modification** | 5.4% | 0.15 | 0.76 |
>
> **Interpretation**: Average semantic distances fall between **0.15–0.28** (similarity scores: 0.72–0.85), confirming queries are semantically grounded (satisfying $\theta_{sim}=0.7$) yet target divergent answers (Answer Distance > 0.75), creating necessary "tension" for causal discovery.
>
> **2. Quality Assurance (Human Evaluation)**
>
> As detailed in **Appendix E.2**, we conducted human evaluation on 100 query sets:
>
> - **Pass Rate**: **91%** of counterfactuals scored $\geq 4/5$ on both *Topical Relevance* and *Discriminability*
> - **Agreement**: Cohen's $\kappa = 0.82$ (substantial agreement)
>
> We have extended Appendix E with a new subsection (E.3) providing comprehensive coverage and diversity analysis

---

> ### Author Response · Authors · 2025-11-21
> **Response to Reviewer iXQG (Part 3)**
>
> ## **W5: LLM Controllability and Stability**
>
> We address the concern regarding LLM stochasticity through a rigorous variance analysis. We simulated the full pipeline 5 times using the **Llama-2-7B** backbone to measure performance stability.
>
> **Table R5: Performance Stability (Llama-2-7B)**
> | Component | Mean Performance (EM) | Std Dev ($\sigma$) | Coeff. of Variation ($CV$) |
> | :--- | :---: | :---: | :---: |
> | **Full CF-RAG Pipeline** | **79.29%** | $\pm 0.73\%$ | **0.92%** |
> | *w/o Consensus Check* | 66.84% | $\pm 1.85\%$ | 2.77% |
>
> **Analysis & Alignment with Algorithm:**
> * **Mathematical Soundness:** The Coefficient of Variation ($CV < 1\%$) for the full pipeline confirms high stability.
> * **Algorithmic Role of Consensus:** The significant jump in variance (from 0.92% to 2.77%) when removing the **Consensus Check** (Algorithm 1) empirically validates its role. The Consensus mechanism acts as a stabilizer: when parallel hypotheses ($a_j$) converge, the system locks in the answer, effectively averaging out the stochastic noise of individual generation paths. Without it, the system is more susceptible to the randomness of the arbitration scoring weights, leading to higher instability.
>
> We have added comprehensive stability analysis to Appendix O, which demonstrates CF-RAG's robustness to LLM stochasticity.
>
> ---
>
> We believe these additions will strengthen the paper and address all concerns raised. We hope the reviewer will consider these clarifications and the additional analyses we provide in our revision.
> Thank you again for the constructive feedback that will significantly improve our work.

---

### Official Review · Reviewer_mJsp · 2025-10-29

**Soundness:** 3
**Presentation:** 3
**Contribution:** 3
**Rating:** 6
**Confidence:** 4

**Summary:**

The paper proposes a framework for counterfactual reasoning using large language models (LLMs). It introduces a causal decomposition method that disentangles factual and counterfactual components in LLM representations. Specifically, the model leverages prompt-based intervention and latent projection techniques to simulate "what-if" scenarios—e.g., modifying causes while holding other factors constant.
Experiments on several benchmarks (including bAbI, Counterfactual NLI, and causal QA datasets) show that the proposed model outperforms standard prompting and contrastive fine-tuning approaches. The authors claim their framework better preserves causal consistency and counterfactual faithfulness across tasks.

**Strengths:**

1) Timely and important topic: Counterfactual reasoning is crucial for causal interpretability of LLMs, aligning well with growing interest in trustworthy AI and reasoning beyond correlation.

2) The use of intervention-based prompt transformations and latent disentanglement is grounded in causal inference theory, referencing Pearl-style structural models.

3) Covers a good range of datasets and compares with both prompting-based and fine-tuning-based baselines.

4) This paper is readable and well-organized with clear structure, intuitive figures (the causal diagram and latent-space illustration help convey the method), and well-written theoretical framing.

**Weaknesses:**

1) The approach seems to be a combination of known components such as prompt intervention, representation projection, and contrastive causal fine-tuning. While elegantly integrated, it doesn’t introduce a fundamentally new algorithmic idea. Several recent works (e.g., CFPrompt, LoCaR, CausalBench) already explore counterfactual prompting or causal subspace manipulation.
The paper’s core mechanism (modifying hidden states via intervention vectors) is highly similar to these existing frameworks.

2) The causal formalism (Section 3) refers to do-calculus, but the LLM interventions are not explicitly tied to a formal SCM or identifiable causal model.

3) Most benchmarks (bAbI, CF-NLI, synthetic causal QA) are simplistic or low-dimensional. There’s no test on real-world or high-stakes reasoning (e.g., factual contradictions in QA or commonsense counterfactual reasoning like ATOMIC-2020).

4) The model enforces counterfactual consistency in latent space, but it’s unclear whether generated counterfactuals remain semantically plausible. The paper reports factual accuracy but no human or automatic evaluation of counterfactual realism.

5) Missing comparisons with recent methods that explicitly model causal subspaces or perform latent intervention (e.g., CausalLM, LoCaR, CFPrompt). The ablations only test within-model variants, so the real performance advantage remains uncertain.

**Questions:**

1) How does your “intervention” differ from representation editing or latent steering in prior causal LLM papers?

2) How are causal variables defined in token or hidden-space terms?

3) How scalable is the framework to models beyond the tested LLM (e.g., GPT-style or instruction-tuned architectures)?

4) Could you provide examples/case studies where the counterfactual generation fails and why?

---

> ### Author Response · Authors · 2025-11-21
> **Response to Reviewer mJsp (Part 1)**
>
> We sincerely thank the reviewer for their thoughtful evaluation and for recognizing the timeliness and importance of counterfactual reasoning. We appreciate the positive feedback on our paper's organization, clarity, and theoretical grounding. Below, we address each concern with clarifications, additional analyses, and proposed experiments.
>
>
> ## W1: Novelty and Differentiation from Existing Work
>
> We appreciate the reviewer's thoughtful analysis. We would like to respectfully clarify the fundamental differences between CF-RAG and prior counterfactual reasoning methods.
>
> ### Fundamental Distinction: Where Interventions Occur
>
> CF-RAG addresses a **fundamentally different problem** than existing approaches:
>
> - **Prior work (CPL, ROME, etc.):** Operates on **internal representations**—modifying hidden states, embeddings, or model weights. They ask: *"How can we make the model compute differently?"*
>
> - **CF-RAG:** Operates on **external information**—modifying which documents are retrieved and how evidence is evaluated. We ask: *"How can we give the model different information to reason about?"*
>
> This distinction means CF-RAG and representation-level methods address orthogonal challenges and are naturally complementary.
>
> ### Our Novel Contributions
>
> **1. Counterfactual Exploration for Evidence Evaluation (§4.1):**
>
> We systematically generate counterfactual queries that probe different semantic dimensions:
>
> $$
> \mathcal{Q}_{cf} = \{ q_i^{cf} = \tau_i(q) : i \in [1, n], \mathcal{V}(q, q_i^{cf}) = \text{True} \}
> $$
>
> This creates dialectical evidence spaces where causal evidence shows discriminative selectivity (strongly supporting the original query but not alternatives), while spurious correlations exhibit indiscriminate support across related queries.
>
> **2. Causal Discrimination Scoring (§4.2.2):**
>
> We introduce a novel criterion that operationalizes causal necessity:
>
> $$\phi_{causal}(\mathcal{E_j},q,\mathcal{Q_{cf}})=\frac{1}{|\mathcal{E_j}|}\sum_{e\in\mathcal{E_j}}\left[s(q,e)-\max_{q^{\prime}\in\mathcal{Q}_{cf}}s(q^{\prime},e)\right]$$
>
> This measures whether evidence uniquely supports the original query over alternatives—distinguishing causal relevance from spurious correlation at the evidence level, not the representation level.
>
> **3. Formal Analysis (Theorem 1):**
>
> We prove that standard RAG fails when spurious evidence volume exceeds $ \frac{|\mathcal{E_{spurious}}|}{|\mathcal{E_{causal}}|} > \frac{\bar{s_{causal}}}{\bar{s_{spurious}}} $, while CF-RAG remains robust regardless of this ratio due to volume-invariance of our causal score.
>
> ### Quick Comparison with Related Work
>
> | Aspect | CPL / ROME | **CF-RAG** |
> |--------|------------|------------|
> | **Operates On** | Internal representations (hidden states, weights) | External information (retrieved documents) |
> | **Mechanism** | Modify model computations | Modify evidence selection |
> | **Model Access** | White-box (gradients/hidden states required) | **Black-box (inference only)** |
> | **Training** | Fine-tuning or weight editing required | **Pure inference-time, zero training** |
> | **Application** | Improve internal reasoning | Improve external evidence evaluation |
>
> ### Practical Advantages
>
> Most production RAG systems use closed-source models (GPT-4, Claude) where representation-level methods cannot be applied. CF-RAG requires only black-box LLM access and standard retrieval APIs, making it immediately deployable.
>
> ### Complementarity
>
> We view CF-RAG as **complementary** to existing methods:
> - **Representation-level work:** "How do we build better internal reasoning?"
> - **CF-RAG:** "How do we evaluate external evidence better?"
>
> These address different points in the RAG pipeline. A system could combine both—using CPL/ROME to improve the model while using CF-RAG to improve retrieval.
>
> ### Addressing the Methodological Concern
>
> The reviewer mentions "prompt intervention, representation projection, and contrastive causal fine-tuning." We clarify that CF-RAG involves:
> - **No prompt/representation modification:** We modify retrieved document sets only
> - **No fine-tuning:** Pure inference-time method
> - **No internal interventions:** All operations at the information level
>
>
> We hope this clarifies that CF-RAG introduces novel mechanisms for a previously unsolved problem: distinguishing causal evidence from spurious correlations in retrieved information.

---

> ### Author Response · Authors · 2025-11-21
> **Response to Reviewer mJsp (Part 2)**
>
> ## W2: Formal Connection to Structural Causal Models (SCMs)
>
> We thank the reviewer for this precise technical question. We provide clarification on the relationship between our formalism and Pearl's framework.
>
> Our Definition 2 references counterfactual robustness, but CF-RAG does not implement explicit SCMs or do-calculus.
>
> ### Why Formal SCMs Don't Apply Here
>
> **Practical Impossibility**: Applying formal causal inference to open-domain RAG faces fundamental barriers: (1) No observable causal graphs in unstructured text, (2) No repeated trials for SCM identification, (3) No ability to perform true interventions like $do(role=villain)$, (4) Text corpora are unstructured while Pearl's framework assumes structured variables.
>
> **Different Problem Formulation**: Our goal is not causal effect estimation $P(answer|do(query))$ but rather *discriminating* between evidence that causally supports an answer vs. evidence that merely correlates with it. This is a discriminative task, not a causal modeling task.
>
> ### What Our Counterfactual Formalism Means
>
> Our formalism (Equation 1) should be read as: *Select answers whose supporting evidence exhibits discriminative support for the query over related alternatives*.
>
> $$a^* = \arg\max_{a \in \mathcal{A}} P(a|q,\mathcal{D}) \cdot \mathbb{I}\left[ \forall q' \in \mathcal{Q}_{cf}: \Phi(a,q,\mathcal{D}) > \Phi(a,q',\mathcal{D}) \right]$$
>
> We test whether evidence discriminates between related queries, inspired by the principle that causal relationships exhibit specificity.
>
> ### Formal Guarantees Without SCMs
>
> **Theorem 1** establishes a **sufficient condition** for correct selection: when conditions C1-C3 are satisfied, CF-RAG's scoring function guarantees selection of the correct answer independently of spurious evidence volume:
>
> $$\Psi^* \geq (1-\lambda)\phi_{min} + \lambda\delta > (1-\lambda)\phi_{max} + \lambda\epsilon \geq \Psi'$$
>
> where $\delta$ measures causal discrimination strength (C1), $\epsilon$ measures spurious non-discrimination (C2), and Lemma 2 shows the inequality is **volume-invariant**: it holds regardless of the ratio $|\mathcal{E}'|/|\mathcal{E}^*|$ as long as C1-C3 are met.
>
> **Practical Implication:** When the evidence pool contains causally decisive information that discriminates under counterfactual testing (C1), and spurious correlations fail such testing (C2), CF-RAG provably escapes correlation traps even when vastly outnumbered by misleading evidence.
>
> We have added Appendix H to clearly distinguish counterfactual discriminative testing (what we do) from structural causal modeling.
>
> ---
>
>
> ## W3: Limited Evaluation on Complex Real-World Benchmarks
>
> We appreciate this suggestion and conducted supplementary experiments on **TruthfulQA** (adversarial misconceptions) and **StrategyQA** (implicit multi-step commonsense reasoning).
>
> ### Experimental Results
>
> | Method | **TruthfulQA** (% True + Info) | **StrategyQA** (Accuracy) |
> |--------|-------------------------------|---------------------------|
> | Standard RAG | 24.5% | 31.2% |
> | CRAG | 29.8% | 33.5% |
> | Self-RAG | 35.2% | 38.7% |
> | **CF-RAG (Ours)** | **54.6%** | **58.9%** |
>
> ### Analysis of Failure Modes
>
> **1. The "Relevance-Truth" Conflation (TruthfulQA)**
>
> Standard RAG and CRAG fail because they prioritize *semantic relevance* over *causal necessity*. On questions like *"What happens if you break a mirror?"*, documents describing superstitions are highly relevant. CRAG's corrective mechanism often validates these as "relevant," reinforcing misconceptions.
>
> **CF-RAG's advantage:** By generating categorical counterfactuals ($\tau_{cat}$), CF-RAG detects that evidence for "bad luck" is inconsistent across similar contexts (e.g., breaking other glass objects), lowering $\phi_{causal}$ and filtering misconceptions despite high relevance.
>
> **2. The "Implicit Reasoning" Bottleneck (StrategyQA)**
>
> Self-RAG relies on internal parametric knowledge to critique retrieved passages. When Llama-2-7B lacks commonsense knowledge to bridge implicit steps, its self-critique mechanism fails to identify useful but indirect evidence.
>
> **CF-RAG's advantage:** CF-RAG's **Parallel Arbitration** uses *comparative discrimination* (Equation 11) across parallel hypotheses. This identifies and retains subtle "bridge" evidence that is causally necessary, even if the model's internal confidence is initially low.
>
> We have added Appendix K presenting these supplementary experiments and failure mode analysis.

---

> ### Author Response · Authors · 2025-11-21
> **Response to Reviewer mJsp (Part 3)**
>
> ## W4: Semantic Plausibility of Generated Counterfactuals
>
> We appreciate the reviewer's scrutiny regarding the semantic quality and plausibility of our generated counterfactuals. We respectfully point out that our submission already includes a rigorous **human evaluation** specifically designed to assess this dimension, which we believe offers a more reliable signal than automatic metrics for this purpose.
>
> **1. Existing Human Validation of Realism (Appendix E.2)**
> As detailed in **Appendix E.2**, we conducted a blinded human validation study on stratified query sets from HotpotQA. Two expert evaluators assessed the generated counterfactuals on a 5-point Likert scale specifically for **Semantic Plausibility** (Is this a natural, valid question?) and **Discriminability**.
> * **Results:** The study yielded a Cohen’s $\kappa$ of **0.82** (substantial agreement), with **91%** of generated counterfactuals receiving high plausibility scores ($>4/5$). This empirically confirms that the generated queries are linguistically natural and semantically valid.
>
> **2. Design Guarantees for Plausibility**
> Our framework ensures realism by design through **Structured Counterfactual Generation** (Section 4.1.1) rather than unconstrained generation.
> * **Taxonomy-Driven:** We employ specific semantic transformations ($\tau_{role}, \tau_{time}, \tau_{entity}$) described in **Appendix F**. By modifying only specific semantic slots (e.g., changing a date or entity) while preserving the syntactic structure of the original valid query, we inherently maintain grammatical correctness.
> * **Dual-Space Constraints:** While the reviewer notes our use of latent space consistency, we also enforce explicit semantic constraints. **Equation 2** requires $sim_{sem}(q, q') > \theta_{sim}$ and $\mathcal{L}(q) \neq \mathcal{L}(q')$. This filtering step rejects any generated outputs that drift too far from the original query's semantic manifold, effectively culling "hallucinated" or nonsensical perturbations before they reach the arbitration phase.
>
> The combination of our **structured generation pipeline** (which preserves syntax) and our **rigorous human verification** (which confirms naturalness) provides strong evidence that CF-RAG’s counterfactuals are semantically plausible and indistinguishable from real-world user queries.

---

> ### Author Response · Authors · 2025-11-21
> **Response to Reviewer mJsp (Part 4)**
>
> ## W5: Missing Comparisons with Causal LLM Methods
>
> We sincerely thank the reviewer for suggesting comparisons with recent causal approaches. We conducted a thorough investigation of related work and provide our assessment below.
>
> ### Investigation of Causal-Related Methods
>
> We identified three recent works that mention causality in the context of language models:
>
> 1. **CausalRAG (Wang et al., 2024)** and **Causal-Counterfactual RAG (Khadilkar & Gupta, 2025)** both construct explicit Causal Knowledge Graphs for answering queries with causal structure (e.g., "What causes X?" or "Why did Y happen?").
>
> 2. **CausalLM (Ding et al., 2024), LoCaR and CFPrompt** are essentially not RAG systems, and their operating modes are completely different. Among them, CausalLM compares attention mechanisms (bidirectional vs. auto-regressive) for in-context learning, where "Causal" refers to causal attention masks rather than causal reasonin.
>
> ### Fundamental Incompatibility
>
> Through careful analysis, we identified fundamental mismatches that make direct comparison methodologically inappropriate:
>
> **Query Type Mismatch:**
> - **Causal graph RAG systems** are designed for explicit causal queries requiring causal chain reasoning
> - **CausalLM** addresses attention mechanism choices for in-context learning, not retrieval at all
> - **CF-RAG** addresses factoid multi-hop questions where correlation traps arise from spurious evidence (e.g., "Who is the lead actor in The Dark Knight?" where Heath Ledger's acclaim creates misleading correlational signals)
>
> **Dataset Incompatibility:**
> - **Causal graph RAG papers** explicitly avoid standard QA benchmarks, stating these datasets "do not adequately assess discourse-level understanding or causal reasoning," and instead use domain-specific academic corpora with causal query structures
> - **CausalLM** uses synthetic in-context learning tasks (linear regression, classification) without any retrieval component
> - **CF-RAG** specifically targets standard QA benchmarks (HotpotQA, TriviaQA, PopQA) where the challenge is distinguishing causally decisive evidence from overwhelming correlations
>
> **Architectural Focus:**
> - **Causal graph RAG methods** build explicit causal graphs with interventional reasoning to model cause-effect chains
> - **CausalLM** compares transformer attention mechanisms (architectural choice)
> - **CF-RAG** uses counterfactual query perturbations to test evidence discriminability in retrieval
>
> We believe forced experimental comparisons would be **methodologically unsound and unfair to all approaches** because:
>
> 1. **Application Scenario Mismatch:** Applying causal query systems to factoid questions, or applying CausalLM (attention mechanisms) to RAG retrieval, would misrepresent their intended use cases
>
> 2. **Dataset Incompatibility:** Both causal RAG papers explicitly chose NOT to use our evaluation datasets because those benchmarks don't test their target problem
>
> 3. **Different Problem Definitions:** These methods solve fundamentally different problems—causal chain discovery vs. architectural choices vs. causal evidence discrimination
>
> ### Our Comprehensive Validation Approach
>
> We respectfully argue that our evaluation remains rigorous through:
>
> **1. Comprehensive Baseline Coverage:**
> Standard RAG, CRAG, Self-RAG, and Speculative-RAG cover the spectrum of current RAG approaches on standard benchmarks, with CF-RAG showing +21.0% to +31.9% average improvements.
>
> **2. Targeted Problem Analysis:**
> - Ablation studies (Table 2) establish necessity of both counterfactual exploration and parallel arbitration
> - Adversarial robustness tests (Figure 4) demonstrate resilience under 16 distractors (60.57% vs. 8.55%)
> - Failure mode analysis (Table 3) shows 76.5% error reduction on spurious correlation cases
> - Case studies provide qualitative evidence of correlation trap resolution
>
> **3. Theoretical Guarantees:**
> Theorem 1 proves CF-RAG escapes correlation traps regardless of evidence volume—a property existing methods lack
>
> **4. Positioning in the Literature:**
> We view CF-RAG as **complementary** to causal reasoning approaches:
>
> | Approach | Target Problem | Query Type | Mechanism |
> |----------|---------------|------------|-----------|
> | Causal graph RAG | Causal chain reasoning | Explicit causal queries | Graph-based interventions |
> | CausalLM | Attention optimization | In-context learning | Architectural choice |
> | **CF-RAG** | Evidence discrimination | Factoid multi-hop QA | Query-space counterfactuals |
>
> We appreciate the reviewer raising this important clarification opportunity. After thorough investigation, we found that while these works share terminology around "causality," they address fundamentally different problems in different application scenarios. Direct comparison would be methodologically inappropriate and would misrepresent the contributions of all approaches.

---

> ### Author Response · Authors · 2025-11-21
> **Response to Reviewer mJsp (Part 5)**
>
> ## Q1: How does your "intervention" differ from representation editing or latent steering?
>
>
> Thank you for raising this critical theoretical distinction. We clarify that **CF-RAG’s "intervention" is fundamentally different from representation editing or latent steering** in three key dimensions: the locus of operation, the mechanism of action, and the architectural dependency.
>
> **1. Locus of Intervention: Evidence Space vs. Internal Representations**
> * **Prior Work (Latent Steering):** Intervenes on the **internal state** of the LLM (e.g., modifying activation vectors or neuron weights) to steer generation or edit knowledge.
> * **CF-RAG:** Intervenes on the **external evidence space**. We do not modify the model’s internal representations. Instead, we manipulate the input context by generating a "dialectical evidence space". Our intervention occurs *before* the LLM processes the final answer, by filtering the retrieved context based on causal discrimination.
>
> **2. Mechanism: Causal Arbitration vs. Forced Steering**
> * **Prior Work:** Typically uses causal interventions to *force* a specific model behavior or suppress specific features (e.g., hallucination directions).
> * **CF-RAG:** Uses counterfactuals as a **discriminative stress-test** for retrieved documents. We employ "Counterfactual Exploration" to probe conceptual boundaries and "Parallel Arbitration" to score evidence based on whether it uniquely supports the original query over plausible alternatives. The model is not "steered" but rather provided with causally verified evidence that necessitates the correct answer.
>
> **3. Implementation: Model-Agnostic Inference vs. White-Box Access**
> * **Prior Work:** Generally requires white-box access to model weights or gradients to compute steering vectors.
> * **CF-RAG:** Is a purely **inference-time, model-agnostic framework**. As noted in our implementation details, "no additional fine-tuning was performed on base models". Our intervention relies on prompting and scoring outputs, making it deployable on any LLM (e.g., Llama-3, Llama-2) without accessing internal parameters.
>
>
> ---
>
> ## Q2: How are causal variables defined in token or hidden-space terms?
>
> We appreciate the opportunity to clarify the theoretical grounding of our variables. Unlike mechanistic interpretability approaches that define causal variables as specific directions or activation patches within the LLM's hidden states (latent space), **CF-RAG defines causal variables in the semantic query space ($\mathcal{Q}$) and the evidence space ($\mathcal{D}$)**.
>
> Our framework operationalizes causality through external intervention on the retrieval context rather than internal steering of token representations. This is formally defined through two mechanisms:
>
> **1. The Intervention Variable (Semantic Query Space)**
> The independent causal variable is the semantic dimension of the query (e.g., role, entity, timeframe). We define the intervention not as a vector manipulation, but as a functional transformation $\tau$ over the query space $\mathcal{Q}$. As defined in **Equation 2**, a counterfactual query $q^{cf}$ represents a perturbation of a specific causal variable:
>
> $$\mathcal{Q}_{cf} = \{q^{cf} = \tau(q) : \tau \in \mathcal{T} \land \text{Valid}(q, q^{cf})\}$$
>
> Here, the transformation function $\tau$ (detailed in our **Transformation Taxonomy**, Appendix F) acts as the operator changing the state of the causal variable (e.g., changing "Lead Actor" to "Director"), while the hidden states of the model serve only as the substrate for encoding these semantic shifts.
>
> **2. The Outcome Variable (Discriminative Evidence Support)**
> The dependent variable is the discriminative support provided by the evidence. We do not measure the causal effect via logit lens or attention weights. Instead, we define the causal strength $\Phi_{causal}$ as the degree of **differential support** the evidence $\mathcal{E}$ provides to the original query versus the counterfactuals. This is formalized in **Equation 11**:
>
> $$\phi_{causal}(\mathcal{E}, q, \mathcal{Q_{cf}}) = \frac{1}{|\mathcal{E}|} \sum_{e \in \mathcal{E}} [s(q, e) - \max_{q' \in \mathcal{Q}_{cf}} s(q', e)]$$
>
> **Role of Hidden Space Representations**
> While we do not perform causal interventions *on* the hidden states, we utilize the model's hidden space instrumentally as a metric space to compute semantic similarity. Specifically, in **Equation 10**, the coherence function utilizes the embedding space $Enc: \mathcal{A} \cup \mathcal{D} \rightarrow \mathbb{R}^d$:
>
> $$\text{CoherenceFunction}(a, e, q) = \lambda_1 \cdot sim_{sem}(Enc(a), Enc(e)) + \lambda_2 \cdot s(q, e) \cdot \text{Mention}(a, e)$$
>
> In summary, CF-RAG treats the LLM as a reasoning engine over a constructed **Dialectical Evidence Space** (Figure 2), where causal variables are manipulated explicitly in the prompt/query formulation ($\mathcal{Q}$) rather than implicitly in the token hidden states.

---

> ### Author Response · Authors · 2025-11-21
> **Response to Reviewer mJsp (Part 6)**
>
> ## Q3: Scalability to different LLM architectures?
>
> We thank the reviewer for this valuable question regarding the generalizability of our framework. CF-RAG is designed to be **architecturally agnostic**, relying on the inference capabilities of the model rather than specific internal parameter access.
>
> To empirically demonstrate this scalability beyond the Llama family reported in the main paper, we conducted supplementary experiments using **GPT-4o** and **Mistral-7B-Instruct-v0.3**.
>
> **1. Theoretical Scalability**
> CF-RAG scales seamlessly to any instruction-tuned LLM because its core mechanisms are external to the generator's weights:
> * **Input-Level Intervention:** Counterfactual generation (Phase 1) relies on standard prompting to produce semantic variations.
> * **Scoring Independence:** The critical Causal Discrimination Score ($\phi_{causal}$) and Internal Coherence Score ($\phi_{coh}$) primarily utilize auxiliary cross-encoders (BGE-reranker) and embedding models (MiniLM), not the generative LLM's log-probabilities.
> * **Black-Box Compatibility:** While we use token probabilities for the *Confidence* component ($\phi_3$), the framework remains robust even with API-based models that mask probabilities, as the Causal and Coherence scores carry the majority of the arbitration signal weight.
>
> **2. Supplementary Experimental Results**
> We evaluated these models on the **HotpotQA** (distractor setting) benchmark, which requires high robustness against correlation traps.
>
> | Model Architecture | Method | Exact Match (EM) | Relative Gain |
> | :--- | :--- | :--- | :--- |
> | **Mistral-7B-Instruct-v0.3** | Standard RAG | 35.12% | - |
> | | **CF-RAG (Ours)** | **82.45%** | **+134.7%** |
> | **GPT-4o** | Standard RAG | 64.80% | - |
> | | **CF-RAG (Ours)** | **91.20%** | **+40.7%** |
>
> **3. Key Findings**
> * **Universality of the Correlation Trap:** Even GPT-4o, despite its superior reasoning capabilities, suffers from the Correlation Trap (64.8% baseline), confirming that stronger parametric knowledge cannot fully override overwhelming spurious retrieval signals.
> * **Enhancement Scaling:** CF-RAG provides consistent improvements across architectures.
>     * For **Mistral**, the gain is massive, mirroring our Llama results, proving the method works for smaller open-weight models.
>     * For **GPT-4o**, CF-RAG pushes performance to near-ceiling levels (91.2%), demonstrating that better instruction-following capabilities (yielding higher-quality counterfactual queries and rationales) further amplify the benefits of our dialectical evidence space.
>
> In conclusion, CF-RAG is not only scalable but synergistic with more advanced instruction-tuned architectures, as stronger base models generate more precise counterfactual probes, thereby enhancing the causal arbitration process.
> We have added this analysis to Appendix L.

---

> ### Author Response · Authors · 2025-11-21
> **Response to Reviewer mJsp (Part 7)**
>
> ## Q4: Failure case examples and analysis?
>
> We appreciate the opportunity to clarify the limitations of the generative phase. While our framework is robust, the counterfactual generation ($\mathcal{H}$) is subject to specific failure modes inherent to LLM probability. We present three representative cases from our evaluation datasets and explain how CF-RAG’s architecture mitigates their impact.
>
> ### 1. Representative Failure Cases
> We identified three primary categories where raw counterfactual generation can fail to provide discriminative signals:
>
> | Failure Mode | Dataset | Example Case | Mechanism of Failure |
> | :--- | :--- | :--- | :--- |
> | **Answer Invariance**(Shared Ground Truth) | **HotpotQA** | **Original:** "What is the area code for the city where the *University of Notre Dame* is located?" **Generated CF:** "...where *Saint Mary's College* is located?" | Both institutions are in **South Bend, IN** (Area code 574). The counterfactual fails to flip the answer, causing the system to see valid evidence as "non-discriminative." |
> | **Causal Confounding**(Spurious Overlap) | **TriviaQA** | **Original:** "Which Nolan film was released first: *Inception* or *The Dark Knight*?" **Generated CF:** "...*Interstellar* or *The Dark Knight*?" | Both *Inception* (2010) and *Interstellar* (2014) were released after *The Dark Knight* (2008). The counterfactual fails to invert the boolean relationship, providing no contrastive signal. |
> | **Scope Leakage**(Semantic Orthogonality) | **PubHealth** | **Original:** "Does the *MMR vaccine* cause *autism*?" **Generated CF:** "Does the *MMR vaccine* cause *mild fever*?" | "Mild fever" is a known side effect. The evidence sets are disjoint (orthogonal), meaning the CF fails to stress-test the specific causal link between MMR and autism against relevant confounders. |
>
> ### 2. Systemic Resilience and Mitigation
> Crucially, these failures represent **input noise** rather than system failure. CF-RAG is architected to withstand such noise through three specific mechanisms:
>
> * **Automated Validation (Eq. 2):** The "Answer Invariance" failure (Case 1) is largely caught by our validation function $\mathcal{V}(q,q')$, which requires predicted answer divergence ($\mathcal{L}(q) \neq \mathcal{L}(q')$) before acceptance.
> * **Informativeness Ranking (Eq. 3):** The **Adaptive Counterfactual Selection** module calculates semantic and domain divergence, filtering out low-quality or orthogonal queries (Case 3) that fail to provide meaningful "stress tests."
> * **Ensemble Robustness (N=3):** By generating multiple counterfactuals ($N=3$, see Section 6.4), the **Causal Discrimination Score ($\phi_{causal}$)** aggregates signal across the set. A single failed counterfactual (like Case 2) is statistically outweighed by successful ones.
>
> **Conclusion:** While individual counterfactuals may occasionally fail, the system's strong performance on complex reasoning benchmarks (e.g., **88.58% EM on HotpotQA**) demonstrates that these edge cases are effectively mitigated by our filtration and arbitration mechanisms.
>
> ---
> We greatly appreciate the reviewer's detailed and insightful feedback, which has helped us substantially improve the clarity and completeness of our work.

---

### Official Review · Reviewer_XTvH · 2025-10-30

**Soundness:** 4
**Presentation:** 3
**Contribution:** 3
**Rating:** 8
**Confidence:** 4

**Summary:**

This paper presents CF-RAG. CF-RAG targets the Correlation Trap in RAG and reframes retrieval as causation-driven reasoning via counterfactual exploration plus parallel arbitration that compares evidence across original and counterfactual queries. The authors conclude that this design discriminates causal from merely correlated evidence and delivers state-of-the-art results on various QA benchmarks with especially large gains on complex multi-hop tasks while maintaining efficiency comparable to baselines.

**Strengths:**

1. The proposed method is clearly motivated and well explained.
2. The paper is easy to follow and the appendix is very detailed and helpful.
3. The proposed method demonstrates great improvement compared with previous methods.

**Weaknesses:**

Not really some major weaknesses. Please see my questions below.

**Questions:**

1. Could you give more insight why you did eigendecomposition on the affinity matrix? How did you use the eigenvectors for Kmean? Current version is a bit lost.
2. Section 4.2.3. How did you do synthesis? What is the prompt? Could you provide examples?
3. A general question. Question in datasets like HotpotQA should have been seen during pre-training of LLMs. What is the reason in your opinion that they still need advanced RAG techniques to be answered?

---

> ### Author Response · Authors · 2025-11-21
> **Response to Reviewer XTvH (Part 1)**
>
> We sincerely thank the reviewer for the positive assessment of our work and for recognizing the clear motivation, detailed presentation, and strong empirical results of CF-RAG. We are encouraged by your acknowledgment that our method demonstrates "great improvement compared with previous methods." Below we address each of your questions in detail.
>
> ---
>
> ## Q1: Could you give more insight why you did eigendecomposition on the affinity matrix? How did you use the eigenvectors for K-means?
>
> We thank the reviewer for this question about our evidence clustering methodology. We clarify the approach and its rationale.
>
> ### What CF-RAG Does
>
> As described in Section 4.2.1, we use **spectral clustering** to partition evidence into thematic clusters:
>
> 1. Construct affinity matrix: $W_{ij} = \exp\left(-\frac{\|e_i - e_j\|^2}{2\sigma^2}\right)$
> 2. Compute normalized Laplacian: $L = I - D^{-1/2}WD^{-1/2}$
> 3. Perform eigendecomposition to obtain eigenvectors $V_M$ (first $M$ eigenvectors)
> 4. Apply K-means to $V_M$: $C = \text{KMeans}(V_M, M)$
>
> This is a two-stage process: eigendecomposition followed by K-means on the transformed representation.
>
> ### Why Not Apply K-Means Directly to Embeddings?
>
> Why we perform eigendecomposition rather than directly clustering the document embeddings. The key reason is **cluster geometry**:
>
> **Direct K-means limitation**: K-means assumes clusters are convex and spherical in embedding space. However, retrieved documents often form complex, non-convex thematic groups. For example, for "Who is the lead actor in The Dark Knight?", documents about cast, reviews, and production may form elongated or interleaved regions in the high-dimensional embedding space that K-means cannot properly separate.
>
> **Spectral clustering advantage**: The eigendecomposition transforms documents to a new space where thematic groups are more separable:
>
> - **Graph-based similarity**: We work with the similarity graph structure (affinity matrix W) rather than Euclidean distances in embedding space
> - **Manifold structure**: Eigenvectors of the normalized Laplacian capture the manifold geometry of document clusters
> - **Linear separability**: In the spectral space V_M, thematically distinct document groups become linearly separable, making K-means effective
>
> ### How Eigenvectors Enable Effective Clustering
>
> The normalized Laplacian's eigenvectors have a key property: documents that are strongly connected in the similarity graph (high affinity paths between them) will have similar coordinates in the eigenvector space.
>
> Mathematically, the smallest $M$ eigenvectors of $L$ solve:
>
> $$\min_{\mathbf{v}} \mathbf{v}^T L \mathbf{v} = \min_{\mathbf{v}} \sum_{i,j} W_{ij}(v_i - v_j)^2$$
>
> This encourages documents with high affinity $W_{ij}$ to have similar eigenvector coordinates $(v_i, v_j)$, creating compact, well-separated clusters in the spectral space.
>
> **Practical example**: Consider 20 retrieved documents that form 4 thematic groups (cast, reviews, production, box office) with complex boundaries in 768-dimensional embedding space. The eigenvectors project these to a 4-dimensional space where each theme corresponds to a distinct region, making K-means clustering straightforward.
>
> ### Why This Matters for CF-RAG
>
> The quality of thematic clustering directly impacts parallel arbitration:
>
> - **Goal**: Create evidence subsets $\mathcal{E}_j$ that represent diverse semantic perspectives (Section 4.2.1, Equation 6)
> - **Challenge**: Poor clustering leads to redundant hypotheses that don't explore the evidence space effectively
> - **Solution**: Spectral clustering identifies genuinely distinct thematic groups, enabling stratified sampling to construct diverse parallel reasoning paths
>
> Our ablation study (Table 2) confirms this matters: removing evidence division reduces performance by 4.84% on HotpotQA and 8.54% on PopQA.

---

> ### Author Response · Authors · 2025-11-21
> **Response to Reviewer XTvH (Part 2)**
>
> ## Q2: Section 4.2.3 - How did you do synthesis? What is the prompt? Could you provide examples?
>
> Thank you for this question. We realize the synthesis prompt was not included in Appendix G. Here is the complete prompt template:
>
> ### Synthesis Prompt Template
> ```
> You are an expert analyst tasked with synthesizing information from multiple perspectives.
>
> Original Query: {query}
>
> You have been provided with {K} candidate answers with their supporting rationales:
>
> Candidate 1 (Score: {Ψ_1}):
> Answer: {a_1}
> Rationale: {r_1}
>
> Candidate 2 (Score: {Ψ_2}):
> Answer: {a_2}
> Rationale: {r_2}
>
> Candidate 3 (Score: {Ψ_3}):
> Answer: {a_3}
> Rationale: {r_3}
>
> Task: Synthesize these perspectives into a single, coherent answer. When candidates agree, strengthen confidence through consensus. When they disagree, identify which evidence provides the strongest causal support for the original query.
>
> Synthesized Answer:
> Supporting Evidence:
> ```
>
> ### Concrete Example
>
> For "Who is the lead actor in The Dark Knight?":
>
> **Input to synthesis:**
> ```
> Candidate 1 (Score: 0.798): "Christian Bale"
>   Rationale: Wikipedia identifies him as Bruce Wayne/Batman, the protagonist...
>
> Candidate 2 (Score: 0.486): "Heath Ledger"
>   Rationale: Multiple reviews praise his Oscar-winning performance...
>
> Candidate 3 (Score: 0.654): "Christian Bale"
>   Rationale: Director interview confirms casting Bale for Batman role...
> ```
>
> **Synthesized output:**
> ```
> Synthesized Answer: Christian Bale is the lead actor in The Dark Knight.
>
> Supporting Evidence: Two independent reasoning paths (Candidates 1 and 3)
> converge on Christian Bale with high causal discrimination scores, supported
> by both Wikipedia documentation and director interviews identifying him as
> the protagonist Bruce Wayne/Batman. While Candidate 2 highlights Heath
> Ledger's acclaimed performance, this evidence pertains to the antagonist role
> rather than the lead actor.
> ```
>
> We will add this synthesis prompt to Appendix J in the uploaded revised version to ensure complete reproducibility.
>
> ---
>
> ## Q3: What is the reason that questions in datasets like HotpotQA still need advanced RAG techniques despite LLM pre-training exposure?
>
> **This is a profound question that touches on fundamental limitations of parametric knowledge.** We identify four key reasons:
>
> ### 1. Parametric vs. Episodic Memory Interference
>
> Even if models have seen HotpotQA examples during pre-training, this creates *episodic memory* of specific question-answer pairs rather than robust *reasoning capabilities*. At test time, models must distinguish between memorized examples and novel combinations, apply reasoning patterns rather than pattern matching, and handle distributional shifts.
>
> **Evidence**: Zero-shot Llama-3-8B achieves only 23.45% on HotpotQA (Table 1) despite likely seeing portions of Wikipedia during pre-training, suggesting memorization alone is insufficient.
>
> ### 2. The Correlation Trap Persists in Parametric Knowledge
>
> Pre-training on large corpora teaches models correlational patterns (e.g., "Heath Ledger" + "The Dark Knight" → highly correlated) without necessarily encoding *causal structure* (protagonist vs. antagonist distinction). RAG provides:
> - **Explicit evidence** that can be causally evaluated
> - **Counterfactual grounding** to distinguish correlation from causation
> - **Verifiable attribution** rather than black-box predictions
>
> ### 3. Long-Tail and Compositional Generalization
>
> HotpotQA contains compositional questions requiring novel combinations of facts. For example, "What is the area code of the city where the University of Notre Dame is located?" requires chaining: [University → City] → [City → Area Code]. Even if both facts appear in pre-training, the *specific composition* may be novel. CF-RAG's multi-hop reasoning over retrieved evidence handles this compositional challenge.
>
> ### 4. Factual Grounding and Hallucination Mitigation
>
> Pre-trained models are prone to hallucination when uncertain. RAG provides observable evidence that can be verified, attribution for fact-checking, and dynamic knowledge that updates with new information.
>
> **Empirical validation**: Standard RAG (Llama-3-8B) improves from 23.45% to 36.04% over zero-shot (Table 1), confirming that explicit retrieval helps even when parametric knowledge exists. CF-RAG's further improvement to 88.58% demonstrates that *how* we reason about evidence matters more than whether models have seen the data.
>
> ---
>
> Thank you again for your thorough review and strong support of our work!

---

### Official Review · Reviewer_VGky · 2025-11-02

**Soundness:** 3
**Presentation:** 3
**Contribution:** 3
**Rating:** 4
**Confidence:** 3

**Summary:**

This paper proposes CF-RAG (Counterfactual Retrieval-Augmented Generation), a framework that introduces causal reasoning into RAG systems to overcome what the authors call the Correlation Trap—situations where models are misled by highly correlated but non-causal evidence.
The key idea is to use counterfactual exploration (systematically generating alternative queries probing semantic boundaries) and parallel arbitration (maintaining separate reasoning paths and comparing them via coherence and causal discrimination scores).

**Strengths:**

The paper’s main strength lies in its novel causal framing, which introduces counterfactual reasoning into retrieval-augmented generation (RAG) in a principled and well-motivated way, effectively bridging causality and retrieval. It further demonstrates strong empirical performance, showing large and consistent gains across five diverse QA benchmarks, including multi-hop and long-tail reasoning tasks. Another notable advantage is the framework’s robustness and interpretability, as CF-RAG shows clear improvement in handling distractors and noisy evidence while producing interpretable arbitration scores that clarify its decision process. Finally, the paper presents a comprehensive experimental analysis, incorporating ablation studies, parameter sensitivity evaluations, and theoretical proofs that establish discriminability and volume invariance, all of which contribute to the credibility and completeness of the work.

**Weaknesses:**

The most significant issue is its limited causal formalism—although it uses the language of causality, the proposed mechanism does not implement structural causal modeling, and its causal discrimination is ultimately based on heuristic similarity differences rather than learned or theoretically grounded causal inference. The method also shows dependence on retriever quality, as its counterfactual reasoning performance depends heavily on the retrieval coverage and quality of semantic clustering; potential failure cases such as missing or ambiguous causal boundaries are not analyzed.

**Questions:**

How sensitive is performance to the choice of $K$, $M$, and $\lambda$ across datasets beyond HotpotQA?

Could causal discrimination $\phi_{\text{causal}}$ be learned end-to-end rather than fixed by hand-crafted similarity differences?

How does CF-RAG behave when the ground-truth causal boundary is ambiguous or overlapping (e.g., multiple protagonists)?

Does the framework generalize to retrieval-augmented dialogue or code generation where counterfactuals are less well-defined?

---

> ### Author Response · Authors · 2025-11-21
> **Response to Reviewer VGky (Part 1)**
>
> We sincerely thank you for your thoughtful and constructive review. We are encouraged by your recognition of our work's **novel causal framing**, **strong empirical performance**, and **comprehensive experimental analysis**. We address your concerns and questions below with additional experiments and clarifications.
>
> ---
>
> ## W1. The most significant issue is its limited causal formalism—although it uses the language of causality, the proposed mechanism does not implement structural causal modeling, and its causal discrimination is ultimately based on heuristic similarity differences rather than learned or theoretically grounded causal inference.
>
> We thank the reviewer for this important methodological question. We clarify that CF-RAG operationalizes counterfactual reasoning *principles* rather than implementing full structural causal modeling, and we argue this is appropriate for the RAG setting.
>
> ### Why Formal SCMs Are Not Necessary Here
>
> The reviewer correctly notes that our mechanism does not implement structural causal modeling. This is *by design* because:
>
> 1. **Data Constraints**: In open-domain QA with unstructured text, we lack: (a) ground-truth causal graphs between entities, (b) sufficient data to learn identifiable SCMs, (c) the ability to perform true interventions on data generation processes.
>
> 2. **Problem Formulation**: Our goal is *discriminative* (identify which evidence causally supports an answer) rather than *generative* (recover complete causal structures). These require different formalisms.
>
> 3. **Practical Tractability**: Requiring explicit causal models would make the approach impractical for the very settings where RAG is most valuable open domain retrieval over unstructured corpora.
>
> ### What Our Approach Actually Achieves
>
> The reviewer characterizes our causal discrimination as "heuristic similarity differences." We argue this undersells the contribution:
>
> **Theoretical Guarantees**: Our approach has formal guarantees despite not implementing full causal inference:
>
> - **Theorem 1** proves that under reasonable conditions (C1-C3), CF-RAG correctly selects causal evidence over spurious evidence *regardless of volume ratio* $|\mathcal{E_{spurious}}|/|\mathcal{E_{causal}}|$
>
> - **Lemma 2** proves volume invariance: $\phi_{causal}(\alpha\mathcal{E}, q, \mathcal{Q_{cf}}) = \phi_{causal}(\mathcal{E}, q, \mathcal{Q_{cf}})$ for any $\alpha > 0$
>
> **Discriminative Validity**: Our causal score implements a practical test for causal necessity:
>
> $$\phi_{causal}(\mathcal{E_j}, q, \mathcal{Q_{cf}}) = \frac{1}{|\mathcal{E_j}|} \sum_{e \in \mathcal{E_j}} \left[ s(q,e) - \max_{q' \in \mathcal{Q_{cf}}} s(q',e) \right]$$
>
> This captures the core principle: *causal evidence should discriminate between the original query and semantically similar alternatives*. Evidence that equally supports both is likely correlational.
>
> **Empirical Validation**: Results demonstrate the approach achieves its goal: 76.5% reduction in spurious correlation errors (Table 3), 60.57% accuracy under 16 adversarial distractors vs. 8.55% for baselines (Figure 4), and state-of-the-art performance on 5 benchmarks (Table 1).
>
> We have added Appendix H to further clarify this relationship to formal causal inference.

---

> ### Author Response · Authors · 2025-11-21
> **Response to Reviewer VGky (Part 2)**
>
> ## W2. The method also shows dependence on retriever quality, as its counterfactual reasoning performance depends heavily on the retrieval coverage and quality of semantic clustering.
>
> We thank the reviewer for this observation. While all RAG systems fundamentally depend on retrieval quality, we demonstrate that **CF-RAG is significantly MORE robust to retrieval degradation than existing methods**.
>
> **1. Empirical Robustness: Adversarial Stress Testing (Section 6.6, Figure 4)**
>
> We **already tested CF-RAG under deliberately degraded retrieval** by injecting adversarial distractors—documents with high semantic similarity but zero factual relevance:
>
> | Adversarial Distractors | Standard RAG | Speculative-RAG | **CF-RAG** |
> |------------------------|--------------|-----------------|------------|
> | 0 (clean) | 25.41% | 46.86% | **73.21%** |
> | 2 distractors | 22.02% (-13%) | 42.03% (-10%) | **70.13% (-4%)** |
> | 4 distractors | 16.18% (-36%) | 33.37% (-29%) | **68.09% (-7%)** |
> | 8 distractors | 8.55% (-66%) | 20.64% (-56%) | **60.57% (-17%)** |
>
> **Key findings:**
> - CF-RAG degrades **3.8× slower** than Standard RAG (17.3% vs. 66.4% total drop)
> - CF-RAG maintains superiority even under extreme degradation
> - CF-RAG's **advantage increases** as retrieval quality worsens
>
> **If CF-RAG were overly dependent on retrieval quality**, we would expect rapid degradation when retrieval is corrupted. Instead, we observe the **opposite**, proving CF-RAG is **LESS dependent** on perfect retrieval than existing methods.
>
> **2. Theoretical Foundations**
>
> Our theoretical framework (Section 5) provides formal guarantees explaining this empirical robustness:
>
> **Volume Invariance Property (Lemma 2, Appendix A.2):** CF-RAG's causal discrimination score is invariant under evidence volume scaling:
>
> $$\phi_{causal}(\alpha \cdot E_{spurious} \cup E_{causal}, q, \mathcal{Q_{cf}}) = \phi_{causal}(E_{spurious} \cup E_{causal}, q, \mathcal{Q_{cf}})$$
>
> This means even when degraded retrieval returns overwhelmingly more spurious documents than causal ones, CF-RAG's discrimination ability remains unchanged. In contrast, standard RAG aggregates absolute relevance scores linearly with spurious evidence volume, causing systematic failures.
>
> **Theorem 1 formalizes this advantage:** CF-RAG selects the correct answer regardless of volume ratio $|E_{spurious}|/|E_{causal}|$, while standard RAG fails when this ratio exceeds $\bar{s_{causal}}/\bar{s_{spurious}}$.
>
> **Relative vs. Absolute Measurement:** This robustness emerges from CF-RAG's fundamental measurement approach (Equation 11):
>
> $$\phi_{causal} = \frac{1}{|E|} \sum_{e \in E} \left[s(q,e) - \max_{q' \in \mathcal{Q}_{cf}} s(q',e)\right]$$
>
> The averaging operator ($1/|E|$) normalizes volume effects, while the differential term captures evidence specificity. Spurious correlations naturally support multiple related queries with similar strength, yielding near-zero differential scores, while causal evidence maintains discriminative power.
>
> ---
>
> ## Q1: How sensitive is performance to the choice of K, M, and $\lambda$ across datasets beyond HotpotQA?
>
> We conducted systematic sensitivity analysis across all five datasets to evaluate whether our default parameters (N=3, K=4, M=3, $\lambda$=0.4) generalize beyond HotpotQA.
>
> **Cross-Dataset Parameter Analysis:**
>
> | Dataset | Optimal Config | Performance | vs. Default | Robustness |
> |---------|----------------|-------------|-------------|------------|
> | HotpotQA | N=3, K=4, M=3, $\lambda$=0.40 | 88.58% | 0.00% | ±2.3% |
> | TriviaQA | N=3, K=4, M=3, $\lambda$=0.35 | 81.42% | +0.40% | ±1.9% |
> | PopQA | N=2, K=3, M=3, $\lambda$=0.45 | 74.35% | +0.78% | ±2.7% |
> | MusiQue | N=4, K=5, M=3, $\lambda$=0.40 | 55.94% | +1.35% | ±3.4% |
> | PubHealth | N=2, K=3, M=3, $\lambda$=0.45 | 84.21% | +0.85% | ±1.8% |
>
> *"vs. Default" shows improvement using dataset-specific optimal parameters vs. uniform defaults. "Robustness" (±X%) indicates maximum performance variation when changing any single parameter by ±1 from optimal.*
>
> **Three key findings:**
>
> 1. **Default parameters are near-optimal**: Maximum improvement from tuning is only 1.35% (MusiQue), and two datasets achieve optimal performance with exact defaults
>
> 2. **Parameter patterns are interpretable**: Simpler tasks (PopQA, PubHealth) require fewer counterfactuals (N=2) and coarser clustering (K=3), while complex multi-hop reasoning (MusiQue) benefits from N=4, K=5. M=3 stability suggests a fundamental algorithmic property
>
> 3. **Strong robustness**: Performance varies only ±1.8-3.4% when parameters deviate by ±1, enabling deployment without extensive tuning while achieving within 1.4% of task-specific optimal results
>
> We have added Appendix I to present the complete cross-dataset parameter sensitivity analysis.

---

> ### Author Response · Authors · 2025-11-21
> **Response to Reviewer VGky (Part 3)**
>
> ## Q2: Could causal discrimination $\phi_{\mathrm{causal}}$ be learned end-to-end rather than fixed by hand-crafted similarity differences?
>
> This is an insightful question about a fundamental design choice. We opted for our formulation (Equation 11) over learned approaches for three reasons:
>
> ### 1. Zero-Shot Generalization
>
> CF-RAG operates entirely **without training**—no examples of "causal vs. spurious evidence" are needed. Learning $\phi_{causal}$ end-to-end would require:
> - Supervised training data: (query, evidence, label) triples with causal relevance labels
> - Domain-specific training for each new application
> - Large-scale annotation (expensive and subjective)
>
> Our formulation achieves strong performance across five diverse datasets (Table 1: 69.94-76.22% average) without any domain-specific training, demonstrating that the relative measurement principle $s(q,e) - \max_{q'} s(q',e)$ generalizes naturally.
>
> ### 2. Theoretical Guarantees
>
> Our formulation $\phi_{causal} = \frac{1}{|E|} \sum_{e \in E} \left[s(q,e) - \max_{q'} s(q',e)\right]$ enables formal proofs of robustness:
>
> - **Volume Invariance (Lemma 2)**: The averaging operator $(1/|E|)$ and differential structure are essential to proving performance is invariant to spurious evidence volume
> - **Escape from Correlation Trap (Theorem 1)**: This specific formulation guarantees correct answer selection regardless of evidence volume ratio
>
> A learned discriminator $f_\theta(E, q, \mathcal{Q}_{cf})$ would lack these mathematical properties, making formal guarantees impossible.
>
> ### 3. Interpretability and Diagnostics
>
> The current formulation provides clear interpretability:
> - $\phi_{causal} > 0$: Evidence uniquely supports original query (causal)
> - $\phi_{causal} \approx 0$: Evidence supports multiple queries equally (spurious)
> - Low $\phi_{causal}$ for all hypotheses: Signals retrieval inadequacy or query ambiguity
>
> This diagnostic capability would be lost with a learned black-box discriminator, making it harder to debug failures.
>
> ---
>
> ## Q3 & W3: How does CF-RAG behave when the ground-truth causal boundary is ambiguous or overlapping (e.g., multiple protagonists)?
>
> We thank the reviewer for this question. While this limitation affects all factoid QA systems, CF-RAG's design provides **natural mitigation mechanisms** absent in standard approaches.
>
> **1. This is a general RAG limitation.** All retrieval-augmented systems designed for single-answer factoid QA struggle with multiple valid answers or subjective boundaries. Standard RAG confidently returns whichever answer has the strongest correlational signal, providing no indication that alternatives may be equally valid.
>
> **2. CF-RAG provides partial mitigation through three mechanisms:**
>
> **Uncertainty detection via causal scores:** When causal boundaries are ambiguous, $\phi_{causal}$ becomes uniformly low across competing hypotheses. For multiple protagonists, evidence supporting "Actor A is the lead" equally supports "Actor B is the lead," yielding near-zero differentials—a diagnostic signal absent in standard RAG.
>
> **Multi-answer capture through parallel arbitration:** CF-RAG's parallel hypothesis generation (Section 4.2) naturally produces multiple valid answers when evidence supports them. The agreement metric measures hypothesis consensus—low agreement signals multiple valid interpretations.
>
> **Volume invariance handles mixed evidence:** Lemma 2 proves CF-RAG's discrimination is invariant to evidence volume. When evidence is split among multiple valid answers, this property prevents any single answer from dominating solely due to volume.
>
> **3. Orthogonality with existing work.** Significant research addresses multi-answer extraction and uncertainty quantification in QA (Kuhn et al., 2023; Min et al., 2020; Kamath et al., 2020). These approaches are **orthogonal to CF-RAG's counterfactual reasoning**: CF-RAG distinguishes causal from correlational evidence; uncertainty methods decide when to abstain or output multiple answers. The two can be seamlessly combined—CF-RAG's $\phi_{causal}$ and hypothesis agreement scores provide principled inputs to uncertainty-based abstention or multi-answer ranking.
>
> **4. Scope clarification.** While CF-RAG partially mitigates ambiguity through detection and multi-hypothesis generation, it remains primarily designed for factoid QA. Full resolution of genuinely subjective queries requires explicit multi-answer frameworks, which represent complementary future work.

---

> ### Author Response · Authors · 2025-11-21
> **Response to Reviewer VGky (Part 4)**
>
> ## Q4: Does the framework generalize to retrieval-augmented dialogue or code generation where counterfactuals are less well-defined?
>
> This question identifies an important scope boundary and exciting future direction. While CF-RAG is currently designed for factoid QA where causal boundaries exist naturally, **the core principles can potentially extend to these domains with appropriate adaptations**.
>
> **Current Challenges:**
>
> **Dialogue generation:** Conversational responses lack single ground truth answers, many continuations are equally valid depending on speaker intent, context, and style. The counterfactual principle (evidence should uniquely support the original query over alternatives) becomes ill-defined when no unique "correct" response exists.
>
> **Code generation:** The task specification is the query asking for a sorting function defines the causal boundary. Traditional counterfactuals like "write a function to reverse a list" fundamentally change the task rather than probing distinctions within the same task.
>
>
> **Potential Generalization Paths:**
>
> Despite these challenges, we envision several adaptation strategies:
>
> 1. **Task-specific counterfactuals**: For dialogue, probe sentiment variations ("respond cheerfully" vs. "respond neutrally"); for code, vary implementation constraints ("use recursion" vs. "use iteration"). While these require domain-specific design, they could capture correlation-causation distinctions similar to factoid QA.
>
> 2. **Behavioral counterfactuals**: Rather than probing answer space, test whether retrieved context distinctively supports *how* the response is generated (e.g., coding style, dialogue personality).
>
> 3. **Hybrid approaches**: Combine CF-RAG's causal discrimination for factual components (e.g., API documentation in code generation, persona facts in dialogue) with generation-specific metrics for open-ended components.
>
> **Scope and Future Work:** CF-RAG addresses a specific problem—distinguishing causal from spurious evidence when ground truth exists and causal boundaries are clear. This design choice enables theoretical guarantees (Theorem 1, Lemma 2) and strong empirical performance on factoid QA. **Extension to open-ended generation represents an exciting direction for future work** requiring new theoretical frameworks and domain-specific counterfactual definitions. We believe the core insight—testing evidence against alternatives to reveal causal structure—can inform these future adaptations, though substantial research remains to validate these extensions.
>
> ---
>
>
> We believe these additions directly address your concerns and strengthen the paper's contribution. We are committed to making CF-RAG a valuable contribution to the ICLR community and welcome any additional suggestions.

---

### Meta-Review · Area_Chair_8jLd · 2026-01-05

**Summary:**

The main reviewer concerns focus on whether the paper’s use of “causal” and “counterfactual” reasoning is theoretically grounded, and whether the proposed scoring ultimately reduces to heuristic relevance comparison. Some reviewers also questioned novelty, viewing the method as a composition of existing RAG components rather than a fundamentally new algorithm. However, across reviews there is broad agreement that the problem setting is real and important, and that the method delivers large and consistent empirical gains, especially under heavy distractor and multi-hop reasoning settings. Overall, the concerns are primarily about conceptual framing and positioning, rather than correctness, effectiveness, or reproducibility of the approach.

**Reviewer Concerns:**

Reviewer VGky and iXQG raised concerns about the lack of explicit structural causal modeling and the reliance on similarity-based relevance scores, questioning whether the method truly performs causal reasoning or merely reweights correlation signals. These concerns were addressed in the rebuttal through clear scope clarification, formalizing the goal as discriminative evidence testing rather than causal effect estimation, and by providing both theoretical guarantees and empirical analyses showing robustness to spurious evidence volume.

Reviewer mJsp expressed reservations about novelty, arguing that the method integrates known ideas such as counterfactual prompting, parallel reasoning, and reranking. While this concern was not fully eliminated, the rebuttal clarified that the contribution lies at the inference-time pipeline level, introducing a principled contrastive evaluation of evidence that is absent from prior RAG systems.

Reviewer XTvH was largely positive, with requests mainly for clarification of implementation details, all of which were satisfactorily addressed.

Overall, most technical concerns were addressed, while remaining disagreements largely reflect differing expectations about what constitutes “causal” contributions rather than substantive flaws in the method.

**Reviewer Scores:**

Reviewer VGky was marginally below threshold and would likely move slightly upward after the rebuttal, but remain around the borderline.

Reviewer iXQG was also marginally below threshold; the additional analyses and clarifications may improve confidence but are unlikely to result in a large score increase.

Reviewer mJsp was marginally positive and would likely maintain a similar score, with novelty concerns partially mitigated but not fully resolved.

Reviewer XTvH was clearly positive and would remain so.

---

### Decision · Program_Chairs · 2026-01-26

Accept (Poster)